# Hierarchical Semantic-Acoustic Modeling via Semi-Discrete Residual Representations for Expressive End-to-End Speech Synthesis

**Yixuan Zhou**[1], **Guoyang Zeng**[2], **Xin Liu**[2], **Xiang Li**[1], **Renjie Yu**[1], **Ziyang Wang**[2],
**Runchuan Ye**[1], **Weiyue Sun**[2], **Jiancheng Gui**[2], **Kehan Li**[1], **Zhiyong Wu**[1],[*] **Zhiyuan Liu**[3]
[1]Shenzhen International Graduate School, Tsinghua University [2]ModelBest Inc
[3]Department of Computer Science and Technology, Tsinghua University
`yx-zhou23@mails.tsinghua.edu.cn`

## Abstract

Generative models for speech synthesis face a fundamental trade-off: discrete tokens ensure stability but sacrifice expressivity, while continuous signals retain acoustic richness but suffer from error accumulation due to task entanglement. This challenge has driven the field towards multi-stage pipelines that rely on pre-trained discrete speech tokenizers, but these create a semantic-acoustic divide, limiting holistic and expressive speech generation. We resolve these dilemma through hierarchical semantic-acoustic modeling with semi-discrete residual representations. Our framework introduces a differentiable quantization bottleneck that induces natural specialization: a Text-Semantic Language Model (TSLM) generates semantic-prosodic plans, while a Residual Acoustic Model (RALM) recovers fine-grained acoustic details. This hierarchical semantic-acoustic representation guides a local diffusion-based decoder to generate high-fidelity speech latents. Critically, the entire architecture is trained end-to-end under a simple diffusion objective, eliminating dependency on external discrete speech tokenizers. Trained on over 1 million hours of speech, our 0.5B-parameter model achieves state-of-the-art zero-shot TTS performance among open-source systems, demonstrating that our approach delivers expressive and stable synthesis. Audio samples are available at: `https://voxcpm.github.io/VoxCPM-demopage/`.

## 1 Introduction

The pursuit of modern text-to-speech (TTS) systems has evolved beyond intelligibility toward the synthesis of genuinely human-like audio, capable of conveying subtle emotions, speaker identity, and contextual nuances (Shen et al., 2018; Ping et al., 2017; Ren et al., 2020; Li et al., 2019). This leap is critical for applications like empathetic virtual assistants and immersive digital avatars, and hinges on a core technical challenge: simultaneously capturing the fine-grained acoustic details that define vocal richness and the long-range semantic structures governing intelligibility and natural prosody.

Inspired by the success of large language models (LLMs), a dominant paradigm frames TTS as a sequence modeling task over discrete tokens from pre-trained neural audio codecs (e.g., EnCodec (Défossez et al., 2022)). Autoregressively or Non-autoregressively predicting these tokens from text or phonemes (Borsos et al., 2023a; Kharitonov et al., 2023; Chen et al., 2025; Wang et al., 2025c; Peng et al., 2024) offers excellent scalability and in-context learning capabilities. However, this approach faces a fundamental "quantization ceiling", as the compression process irreversibly discards subtle acoustic details. To mitigate this quality loss, state-of-the-art TTS systems (Du et al., 2024a;b; 2025; Zhou et al., 2025; Casanova et al., 2024) adopt multi-stage hybrid pipelines. Here, an LLM generates discrete tokens which condition a separate diffusion-based decoder. While improving fidelity, this solution creates a stark semantic-acoustic divide: the LLM operates in an abstract, discrete space unaware of acoustic reality, while the diffusion model performs local refinement with-

---

[*]Corresponding author.

out high-level context. This fragmentation prevents end-to-end optimization and limits holistic and expressive speech synthesis.

Alternatively, other approaches directly model continuous speech representations to avoid quantization loss. Early systems like Tacotron 2 (Shen et al., 2018) and more recent models such as MELLE (Meng et al., 2024) generate mel-spectrograms autoregressively. However, predicting continuous targets under standard regression losses often yields over-smoothed and low-diversity outputs. To address this, recent innovations have explored replacing the regression objective with a denoising process to model the distribution of the next continuous representations, spanning both non-autoregressive paradigms (Shen et al., 2023; Le et al., 2023; Chen et al., 2024) and autoregressive methods(Li et al., 2024; Jia et al., 2025; Peng et al., 2025). Among these, autoregressive approaches have often demonstrated superior performance in capturing natural prosody and expressive variation. This innovation successfully enhances the detail and diversity of generated continuous representations. However, a more fundamental issue persists: in a fully continuous autoregressive model, the tasks of high-level semantic-prosodic planning and fine-grained acoustic rendering are conflated within a single learning objective. The model is forced to simultaneously solve two disparate tasks—requiring different inductive biases—in a continuous output space. This entanglement presents a significant challenge to the modeling capacity of a single LLM, as it must learn to be both a global planner and a local renderer without an inherent architectural bias to separate these functions. We argue that this conflation is a root cause of instability. The model's focus is inevitably pulled towards fitting low-level acoustic textures, which compromises its ability to maintain high-level semantic coherence, leading to the well-known problem of error accumulation over long sequences (Pasini et al., 2024).

In this work, we introduce a unified, end-to-end framework that resolves this trade-off through hierarchical semantic-acoustic modeling with semi-discrete residual representations. Our key insight is that holistic and expressive speech synthesis requires explicit architectural separation between semantic-prosodic planning and acoustic rendering, yet should remain within a cohesive, end-to-end trainable system. The core innovation is a differentiable Finite Scalar Quantization (FSQ) (Mentzer et al., 2024) bottleneck that induces natural specialization: (1) a Text-Semantic Language Model (TSLM) generates semantic-prosodic plans stabilized through quantization, focusing on linguistically meaningful patterns; and (2) a Residual Acoustic Language Model (RALM) recovers fine-grained details lost during quantization, specializing in acoustic refinement. This hierarchical design enables each component to excel at its respective role while maintaining differentiability, and both of them will be used to guide a local diffusion decoder to generate high-fidelity speech latents. Critically, the entire hierarchical model is trained end-to-end under a simple diffusion objective, seamlessly integrating planning and rendering without pre-trained tokenizers. Our main contributions are as follows.

- We propose an end-to-end hierarchical architecture that introduces an internal semi-discrete bottleneck to resolve the expressivity-stability trade-off. This mechanism implicitly addresses task entanglement in continuous models by inducing a beneficial separation between semantic-prosodic planning and fine-grained acoustic modeling within a single, unified framework.

- We introduce a residual learning strategy that, in conjunction with the bottleneck, enables a holistic yet specialized modeling process. Unlike fragmented multi-stage pipelines, our approach achieves functional separation without architectural fragmentation, simplifying the training pipeline and eliminating dependency on external discrete speech tokenizers.

- We demonstrate the efficacy of our approach through large-scale training on over 1 million hours of bilingual speech. The resulting model, VoxCPM-0.5B, achieves state-of-the-art zero-shot TTS performance among open-source systems, validating its practical strength.

- We provide extensive ablation studies that conclusively validate the semi-discrete residual representations as the crucial component for robust, expressive, and long-form synthesis. We will release code and models to support future research.

## 2 RELATED WORK

### 2.1 DISCRETE TOKEN-BASED TTS

The discrete token paradigm has emerged as a dominant approach in modern TTS, leveraging the success of large language models. This method converts speech into discrete representations using

neural audio codecs such as EnCodec (Défossez et al., 2022) and DAC (Kumar et al., 2023) through residual vector quantization (RVQ). AudioLM (Borsos et al., 2023a) and VALL-E (Chen et al., 2025) pioneered this direction by framing TTS as an autoregressive sequence prediction task over discrete acoustic tokens. Subsequent developments include SoundStorm (Borsos et al., 2023b), which introduced non-autoregressive generation for improved efficiency, and Spear-TTS (Kharitonov et al., 2023), which focused on multilingual capabilities with minimum supervision. VoiceCraft (Peng et al., 2024) and XTTS (Casanova et al., 2024) further advanced zero-shot TTS with in-context learning.

Recent advancements have focused on enhancing the scalability, controllability and zero-shot adaptation. CosyVoice (Du et al., 2024a) proposed supervised semantic tokens for improved zero-shot performance, while its successors, CosyVoice 2 and 3 (Du et al., 2024b; 2025) incorporated text-based LLM initialization and streaming synthesis for human-parity quality and low latency. IndexTTS (Deng et al., 2025) and IndexTTS2 (Zhou et al., 2025) introduced precise duration and emotion control in autoregressive token generation, enabling applications with strict timing and expressivity requirements. SparkTTS (Wang et al., 2025b) utilized single-stream decoupled speech tokens for modeling efficiency, and FireRedTTS (Guo et al., 2024) along with its update FireRedTTS-2 (Xie et al., 2025) established frameworks for industry-level generative speech, including long-form multi-speaker dialogue. Despite these progresses, discrete approaches suffer from inherent quantization artifacts, limiting acoustic fidelity and prompting hybrid solutions.

## 2.2    CONTINUOUS REPRESENTATION TTS

To circumvent quantization losses in discrete models, continuous representation approaches directly model speech features such as mel-spectrograms or audio latents. Early systems like Tacotron 2 (Shen et al., 2018) established the encoder-decoder framework for text-to-mel mapping, while FastSpeech (Ren et al., 2020) introduced explicit duration modeling for alignment stability. Recent developments have integrated diffusion processes to enhance detail and diversity. Non-autoregressive models like NaturalSpeech 2 (Shen et al., 2023) and VoiceBox (Le et al., 2023) apply diffusion directly on continuous representations. F5-TTS (Chen et al., 2024) advanced flow-matching for efficient synthesis. Autoregressive paradigms, often superior in prosody and variation, include MELLE (Meng et al., 2024) for mel-spectrogram generation. Innovations like ARDiT (Li et al., 2024) use an autogressive diffusion transformer for TTS, unifying semantic coherence and acoustic naturalness via parameter sharing. DiTAR (Jia et al., 2025) extended this with a patch-based design: a causal LM for inter-patch stability and a bidirectional local diffusion transformer for intra-patch refinement. VibeVoice (Peng et al., 2025) employed next-token diffusion for long-form multi-speaker synthesis. More recent models such as CLEAR (Wu et al., 2025) and FELLE (Wang et al., 2025a) focus on latent autoregressive modeling with token-wise coarse-to-fine hierarchies, while MELA-TTS (An et al., 2025) and KALL-E (Zhu et al., 2024) combine joint transformer-diffusion with next-distribution prediction for improved efficiency and quality. Despite these advances, continuous models often entangle high-level semantic planning with low-level acoustic rendering, leading to instability in long sequences without explicit separation.

## 2.3    HIERARCHICAL AND RESIDUAL MODELING IN TTS

Hierarchical and residual approaches decompose TTS into layered tasks to balance stability and expressivity. HierSpeech++ (Lee et al., 2025) employed variational inference for semantic-acoustic mapping. HALL-E (Nishimura et al., 2025) uses hierarchical neural codecs with LLMs for minute-long synthesis. MARS6 (Baas et al., 2025) builds robust encoder-decoder transformers with hierarchical tokens. DiffStyleTTS (Liu et al., 2024) applies diffusion for hierarchical prosody modeling. HAM-TTS (Wang et al., 2024) introduces hierarchical acoustic modeling with data augmentation for zero-shot TTS. QTTS (Han et al., 2025) features hierarchical parallel architectures for residually quantized codes. These methods address flaws in prior paradigms: implicit designs lack regulated bottlenecks, tokenizer-dependent models suffer discrete losses, and fragmented stages hinder end-to-end optimization. However, few fully integrate explicit residual designs with semi-discrete bottlenecks in a unified framework, as proposed in our work, to achieve implicit disentanglement without external dependencies.

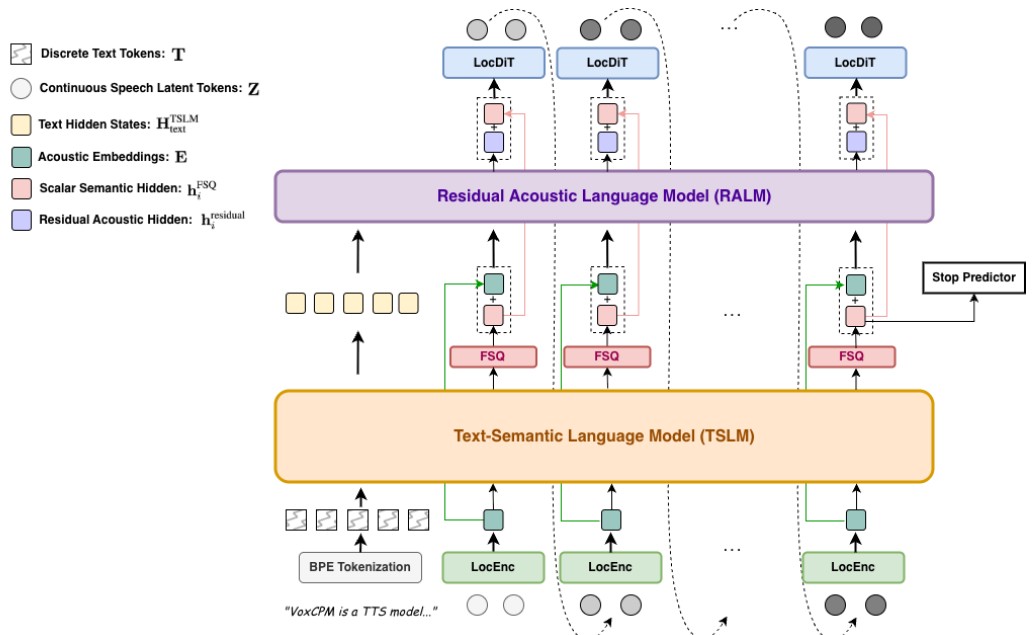

Figure 1: Overall architecture of VoxCPM. The model hierarchically generates speech by first processing audio latents through a LocEnc, then producing a semi-discrete speech skeleton with the TSLM and FSQ, refining acoustic details with the RALM, and finally generating high-fidelity latent output with the LocDiT.

## 3 METHODOLOGY

### 3.1 CORE DESIGN MOTIVATION

Generative speech synthesis faces a fundamental tension between expressivity and stability. Discrete tokenization methods (e.g., discrete speech tokenizers with language models) ensure stable autoregressive generation but irreversibly discard fine-grained acoustic details through quantization. Continuous approaches preserve full fidelity but suffer from error accumulation in long sequences due to information entanglement, often leading to catastrophic failure in intelligibility.

Critically, we identify a key limitation in existing discrete tokenization approaches: methods that directly use FSQ or VQ to obtain discrete codebooks for language modeling face an inherent scalability challenge. As the dimensionality increases to capture richer acoustic information, the codebook size grows exponentially, creating an unmanageably large and sparse vocabulary that language models struggle to predict accurately.

We hypothesize that an effective solution should **structurally separate** the modeling of stable semantic-prosodic content from fine-grained acoustic details while maintaining differentiability for end-to-end training. Our key insight is to introduce a **differentiable quantization bottleneck** that naturally induces this separation through scalar quantization, splitting information into a discrete-like skeleton for content stability and continuous residual components for detail expressivity.

Unlike multi-stage TTS systems composed of seperate LM and diffusion that treat quantization as a means to obtain discrete prediction targets, our approach uses quantization solely as a regularization mechanism to constrain the hidden state space. This distinction allows us to avoid the vocabulary explosion problem while still benefiting from the stabilizing effects of discrete representations.

### 3.2 MODEL OVERVIEW

VoxCPM employs a hierarchical autoregressive architecture that generates sequences of continuous speech latents $\mathbf{Z} = \{\mathbf{z}_1, ..., \mathbf{z}_M\}$ conditioned on input text tokens $\mathbf{T} = \{t_1, ..., t_N\}$, where each $\mathbf{z}_i \in \mathbb{R}^{P \times D}$ represents a patch of $P$ frames with $D$-dimensional VAE latent vectors. The generation

process follows:

$$p(\mathbf{Z}|\mathbf{T}) = \prod_{i=1}^{M} p(\mathbf{z}_i|\mathbf{T}, \mathbf{Z}_{<i}) \tag{1}$$

The core innovation lies in our hierarchical conditioning mechanism with residual representation learning. It is made up of a local audio encoder (LocEnc), a text-semantic language model (TSLM), a residual acoustic language model (RALM) and a local diffusion transformer decoder (LocDiT). A stop predictor is attached to the output of the TSLM to determine the endpoint of generation. As shown in Figure 1, each patch generation involves:

$$\mathbf{z}_i \sim \text{LocDiT}(\mathbf{h}_i^{\text{final}}), \quad \mathbf{h}_i^{\text{final}} = \underbrace{\text{FSQ}(\text{TSLM}(\mathbf{T}, \mathbf{E}_{<i}))}_{\text{stable skeleton}} + \underbrace{\text{RALM}(\cdot)}_{\text{residual details}} \tag{2}$$

where $\mathbf{E}_{<i} = \text{LocEnc}(\mathbf{Z}_{<i})$ represents historical audio context aggregated by a lightweight LocEnc that compresses VAE latent patches into compact acoustic embeddings. The hierarchical backbone produces a conditioning signal $\mathbf{h}_i^{\text{final}}$ that encapsulates both semantic content from TSLM (with FSQ) and acoustic details from RALM. This signal guides the LocDiT to generate the current latent patch $z_i$ through a denoising diffusion process. The entire model is trained end-to-end with gradients flowing through all components, including the FSQ bottleneck via straight-through estimation, ensuring coordinated optimization toward holistic speech synthesis.

### 3.3 HIERARCHICAL SEMANTIC-ACOUSTIC MODELING

Our hierarchical modeling approach is designed to implicitly separate semantic-prosodic planning from fine-grained acoustic synthesis, addressing the fundamental stability-expressivity trade-off through structured representation learning.

#### 3.3.1 TEXT-SEMANTIC LANGUAGE MODEL (TSLM)

The Text-Semantic Language Model forms the main part of our hierarchical architecture, responsible for capturing high-level linguistic structure and generating contextually appropriate speech patterns. Unlike conventional TTS systems that typically operate on phoneme sequences, our approach leverages a pre-trained text language model (MiniCPM-4 (Team et al., 2025)) as its initial backbone, enabling richer contextual understanding and more natural prosody prediction directly from raw text. By processing both text tokens and historical audio context, the TSLM learns to generate semantic content and prosodic structure that evolve naturally throughout an utterance, reflecting the underlying linguistic meaning rather than simply mapping phonemes to acoustic features. The TSLM produces continuous semantic-prosodic representations that encode both the content to be spoken and how it should be prosodically realized, serving as input to the subsequent quantization stage.

#### 3.3.2 SEMI-DISCRETE REPRESENTATION LEARNING VIA FSQ

At the core of our approach lies the Finite Scalar Quantization (FSQ) layer, which projects the continuous hidden states from the TSLM onto a structured lattice to create a semi-discrete representation. The FSQ operation transforms each dimension of the continuous vector through a deterministic scalar quantization:

$$\mathbf{h}_{i,j}^{\text{FSQ}} = \Delta \cdot \text{clip}\left(\text{round}\left(\frac{\mathbf{h}_{i,j}^{\text{TSLM}}}{\Delta}\right), -L, L\right) \tag{3}$$

where $\Delta$ is the quantization step size, $L$ is the clipping range, and round maps values to discrete levels. This transformation creates a structured discrete representation while maintaining differentiability through the straight-through estimator during backward passes.

The FSQ layer acts as a bottleneck, analogous to the first layer of Residual Vector Quantization (RVQ), which captures a coarse semantic-prosodic skeleton (e.g., content, intonation patterns). We term this representation "semi-discrete" as it employs a significantly larger dimensionality than standard FSQ to ensure sufficient informational capacity. Unlike RVQ, where the first layer is a prediction target and subsequent layers model finer details, our FSQ bottleneck serves as an intermediate,

differentiable inductive bias within the continuous data flow. It encourages the model to prioritize modeling stable, high-level components (the semantic-prosodic skeleton) by providing a clear learning signal for what information should be preserved through the bottleneck. This structured approach mitigates error accumulation by reducing the modeling burden on the TSLM, allowing it to focus on the major components of the speech.

### 3.3.3 RESIDUAL ACOUSTIC MODELING

To recover the fine-grained acoustic information attenuated by quantization, we introduce the Residual Acoustic Language Model (RALM). This module specializes in reconstructing those subtle vocal characteristics that conventional discrete methods sacrifice for stability. It processes the quantization residuals along with contextual information to recover speaker identity, spectral fine structure, and micro-prosodic variations:

$$\mathbf{h}_i^{\text{residual}} = \text{RALM}(\mathbf{H}_{\text{text}}^{\text{TSLM}}, \mathbf{H}_{<i}^{\text{FSQ}} \oplus \mathbf{E}_{<i}) \tag{4}$$

Here, the RALM conditions its predictions on both the TSLM hidden states of the text part $\mathbf{H}_{\text{text}}^{\text{TSLM}}$, the semi-discrete representation of speech part $\mathbf{H}_{<i}^{\text{FSQ}}$, and the historical acoustic embeddings $\mathbf{E}_{<i}$. This residual learning approach creates a natural division of labor: the TSLM+FSQ pathway focuses on content stability and prosodic coherence, while the RALM pathway specializes in acoustic expressivity and speaker characteristics.

The final combined representation $\mathbf{h}_i^{\text{final}} = \mathbf{h}_i^{\text{FSQ}} + \mathbf{h}_i^{\text{residual}}$ thus encapsulates both semantic stability and acoustic expressivity, creating a comprehensive signal that guides the subsequent local diffusion process.

### 3.3.4 LOCAL DIFFUSION TRANSFORMER DECODER

The Local Diffusion Transformer (LocDiT) serves as our high-fidelity synthesis module, generating continuous latent patches conditioned on the hierarchical representation $\mathbf{h}_i^{\text{final}}$ produced by the preceding modules. Following DiTAR (Jia et al., 2025), we employ a bidirectional Transformer architecture that enables full receptive field modeling within each patch. To enhance generation consistency, we incorporate the previous patch $\mathbf{z}_{i-1}$ as additional conditioning context, which has been empirically validated to significantly improve output quality by framing the task as outpainting rather than independent patch generation. Besides, we mask the LM guidance in LocDiT condition with a specific probability ratio, for enabling classifier-free guidance (CFG) during inference.

## 3.4 TRAINING OBJECTIVE

The entire model is trained end-to-end using a flow-matching objective that directly optimizes the quality of the generated speech latents. We adopt the conditional flow-matching formulation for its training stability and sampling efficiency:

$$\mathcal{L}_{\text{FM}} = \mathbb{E}t, \mathbf{z}_i^0, \boldsymbol{\epsilon} \left[ |\mathbf{v}_\theta(\mathbf{z}_i^t, t, \mathbf{h}_i^{\text{final}}, \mathbf{z}_{i-1}) - \frac{d}{dt}(\alpha_t \mathbf{z}_i^0 + \sigma_t \boldsymbol{\epsilon})|^2 \right] \tag{5}$$

where $\mathbf{z}_i^t = \alpha_t \mathbf{z}_i^0 + \sigma_t \boldsymbol{\epsilon}$ is the noisy latent at time $t$, with $\boldsymbol{\epsilon} \sim \mathcal{N}(0, I)$, and $\mathbf{v}_\theta$ is the velocity field predicted by the LocDiT.

Simultaneously, a binary classification loss is applied to train the model to predict the end of a speech sequence:

$$\mathcal{L}_{\text{Stop}} = \mathbb{E}_{i \sim \text{sequence}} \left[ \text{BCE} \left( s_\theta(\mathbf{h}_i^{\text{FSQ}}), \mathbb{1}[\text{token } i \text{ is the last}] \right) \right] \tag{6}$$

where $s_\theta$ is a stop-logit projection layer, and BCE denotes the binary cross-entropy loss.

The gradients from this loss are backpropagated through the entire autoregressive hierarchy, including the FSQ layer (via straight-through estimation), the TSLM and the LocEnc. This end-to-end optimization under the combined objective $\mathcal{L} = \mathcal{L}_{\text{FM}} + \lambda \mathcal{L}_{\text{Stop}}$ allows each component to learn its specialized role—semantic planning, stabilization, and acoustic refinement—in a coordinated manner, guided by the unified objective of accurately modeling the continuous speech latents.

## 3.5 Causal Audio VAE

To enable efficient streaming synthesis, we employ a causal Variational Autoencoder that operates in a computationally efficient latent space. VAE is pre-trained separately using a composite objective that combines reconstruction loss in the Mel-spectrogram domain, adversarial training with multi-period and multi-scale discriminators, and a minimal KL-divergence term to regularize the latent space. The use of a latent space rather than raw audio waveforms significantly reduces computational requirements while preserving perceptual quality. The causal nature of the VAE ensures that both encoding and decoding operations can be performed in a streaming fashion, making the entire system suitable for real-time applications where low latency is critical. The detailed implementation of AudioVAE can be found in Appendix D.

## 4 Experiments and Results

### 4.1 Experimental Setup

**Datasets** We conducted experiments on two primary datasets: 1) **Large-scale Bilingual Corpus**: To explore the best performance, we collected an internal large-scale, bilingual dataset totaling over 1 million hours, mainly comprising of Chinese and English speech. 2) **Emilia Dataset**: For comparisons and ablation studies, we used the publicly available Emilia dataset (He et al., 2024) (95K hours). All audio was resampled to 16kHz mono, processed with source separation, voice activity detection (VAD), and automatic speech recognition (ASR) system to obtain text-audio alignment.

**Implementation Details** We implemented VoxCPM using the Megatron framework, with a 0.5B-parameter configuration, comprising a 24-layer Text-Semantic Language Model (TSLM), initialized from the pre-trained MiniCPM-4-0.5B (Team et al., 2025)[1], and a randomly initialized 6-layer Residual Acoustic Language Model (RALM). We trained two models for comparisons: 1) **VoxCPM** was trained with internal large-scale bilingual corpus for 500K steps using 40 NVIDIA H100 GPUs; 2) **VoxCPM-Emilia** was trained on the Emilia dataset for 200K steps using 24 H100 GPUs. Both used the AdamW optimizer with a peak learning rate of $1 \times 10^{-4}$ and a Warmup-Stable-Decay (WSD) schedule (Hu et al., 2024). All ablation studies followed the same 200K-step training protocol on 8 H100 GPUs using the Emilia dataset, employing a fixed learning rate (i.e., without the WSD schedule) of $1 \times 10^{-4}$ to ensure a consistent comparison.

**Evaluation Metrics and Benchmarks** We employed comprehensive subjective and objective evaluations. Objective metrics included Word / Character Error Rate (WER / CER) for intelligibility, speaker embedding cosine similarity (SIM) for voice cloning, and DNSMOS for overall quality. Subjective evaluation involved Mean Opinion Score (MOS) tests rated by 20 native speakers on naturalness (N-MOS) and speaker similarity (S-MOS) using 5-point scales. Models were assessed on two challenging benchmarks: 1) **SEED-TTS-EVAL**, focusing on general TTS intelligibility and similarity in English and Chinese, including a "Hard" set with complex sentences; 2) **CV3-EVAL**, derived from CosyVoice 3 competition, emphasizing expressive and in-the-wild voice cloning.

**Baselines** We compared VoxCPM against a wide range of state-of-the-art open-source TTS systems, including CosyVoice series (Du et al., 2024a;b), MaskGCT (Wang et al., 2025c), F5-TTS (Chen et al., 2024), SparkTTS (Wang et al., 2025b), FireRedTTS series (Guo et al., 2024; Xie et al., 2025), IndexTTS 2 (Zhou et al., 2025), HiggsAudio v2 and so on. All baseline results were obtained using official implementations with default settings, or as reported in their original papers. Details of compared models see Appendix E.

### 4.2 Main Results: Comparison with State-of-the-Art TTS

As shown in Table 1, VoxCPM achieves state-of-the-art performance among open-source models on the SEED-TTS-EVAL benchmark. It attains an English WER of 1.85% and a Chinese CER of 0.93%, surpassing strong competitors like IndexTTS2 and CosyVoice2. Concurrently, VoxCPM maintains high speaker similarity, with SIM scores of 72.9% (EN) and 77.2% (ZH). This demonstrates that the proposed semi-discrete bottleneck effectively balances intelligibility and expressivity by hierarchical semantic-acoustic modeling, mitigating the instability common in continuous models while preserving details often lost in discrete models. The VoxCPM-Emilia variant, trained on a

---

[1] `https://huggingface.co/openbmb/MiniCPM4-0.5B`

Table 1: Performance on Seed-TTS-eval Benchmark

| Model | Params | Data/hrs | EN | | ZH | | Hard | |
|---|---|---|---|---|---|---|---|---|
| | | | WER ↓ | SIM ↑ | CER ↓ | SIM ↑ | CER ↓ | SIM ↑ |
| F5-TTS | 0.3B | 100K | 2.00 | 67.0 | 1.53 | 76.0 | 8.67 | 71.3 |
| MaskGCT | 1B | 100K | 2.62 | 71.7 | 2.27 | **77.4** | - | - |
| CosyVoice | 0.3B | 170K | 4.29 | 60.9 | 3.63 | 72.3 | 11.75 | 70.9 |
| CosyVoice2 | 0.5B | 170K | 3.09 | 65.9 | 1.38 | 75.7 | **6.83** | 72.4 |
| SparkTTS | 0.5B | 100K | 3.14 | 57.3 | 1.54 | 66.0 | - | - |
| FireRedTTS | 0.5B | 248K | 3.82 | 46.0 | 1.51 | 63.5 | 17.45 | 62.1 |
| FireRedTTS-2 | - | 1.4M | 1.95 | 66.5 | 1.14 | 73.6 | - | - |
| Qwen2.5-Omni | 7B | - | 2.72 | 63.2 | 1.70 | 75.2 | 7.97 | 74.7 |
| OpenAudio-s1-mini | 0.5B | 2M | 1.94 | 55.0 | 1.18 | 68.5 | 23.37 | 64.3 |
| IndexTTS 2 | 1.5B | 55K | 2.23 | 70.6 | 1.03 | 76.5 | 7.12 | **75.5** |
| VibeVoice | 1.5B | - | 3.04 | 68.9 | 1.16 | 74.4 | - | - |
| HiggsAudio-v2 | 3B | 10M | 2.44 | 67.7 | 1.50 | 74.0 | 55.07 | 65.6 |
| **VoxCPM-Emilia** | 0.5B | 100K | 2.34 | 68.1 | 1.11 | 74.0 | 12.46 | 69.8 |
| **VoxCPM** | 0.5B | 1.8M | **1.85** | **72.9** | **0.93** | 77.2 | 8.87 | 73.0 |

Table 2: Performance on CV3-eval Benchmark. *denotes close-sourced systems.

| Model | CV3-EVAL | | CV3-Hard-ZH | | | CV3-Hard-EN | | |
|---|---|---|---|---|---|---|---|---|
| | ZH-CER ↓ | EN-WER ↓ | CER ↓ | SIM ↑ | DNSMOS↑ | WER ↓ | SIM ↑ | DNSMOS↑ |
| F5-TTS | 5.47 | 8.90 | - | - | - | - | - | - |
| SparkTTS | 5.15 | 11.0 | - | - | - | - | - | - |
| GPT-Sovits | 7.34 | 12.5 | - | - | - | - | - | - |
| CosyVoice2 | 4.08 | 6.32 | 12.58 | 72.6 | **3.81** | 11.96 | 66.7 | **3.95** |
| OpenAudio-s1-mini | 4.00 | 5.54 | 18.1 | 58.2 | 3.77 | 12.4 | 55.7 | 3.89 |
| IndexTTS2 | 3.58 | 4.45 | 12.8 | **74.6** | 3.65 | 8.78 | **74.5** | 3.80 |
| HiggsAudio-v2 | 9.54 | 7.89 | 41.0 | 60.2 | 3.39 | 10.3 | 61.8 | 3.68 |
| CosyVoice3-0.5B* | 3.89 | 5.24 | 14.15 | 78.6 | 3.75 | 9.04 | 75.9 | 3.92 |
| CosyVoice3-1.5B* | 3.91 | 4.99 | 9.77 | 78.5 | 3.79 | 10.55 | 76.1 | 3.95 |
| **VoxCPM-Emilia** | 4.47 | 5.23 | 22.2 | 62.6 | 3.47 | 10.00 | 62.6 | 3.68 |
| **VoxCPM** | **3.40** | **4.04** | **12.9** | 66.1 | 3.59 | **7.89** | 64.3 | 3.74 |

smaller public dataset, delivers competitive results (EN-WER: 2.34%, ZH-CER: 1.11%). When compared VoxCPM-Emilia against competitive AR models trained on similar data scales (e.g., CosyVoice2, SparkTTS, FireRedTTS), our model consistently outperforms them in stability and similarity. While some NAR models like MaskGCT achieve higher objective metrics, AR models typically excel in prosodic naturalness and expressiveness in the subjective evaluations (as shown in Table 3). This highlights the data efficiency and architectural robustness of our approach, as the FSQ bottleneck stabilizes the learning of semantic-acoustic representations even with less training data. Notably, while DiTAR's phoneme-based approach shows slightly better stability, VoxCPM's use of BPE tokens with pre-trained LLM initialization provides superior text understanding capabilities and eliminates dependency on external phonemizers. Besides, our hierarchical design with residual acoustic modeling reduces the fundamental limitation of direct continuous token modeling, as evidenced in ablation studies.

On the CV3-EVAL benchmark (Table 2), designed to evaluate expressive and in-the-wild performance, VoxCPM excels with a ZH-CER of 3.40% and an EN-WER of 4.04%. Its robustness is further confirmed on the challenging CV3 Hard-Test set, where it achieves an EN-WER of 7.89%, outperforming even close-sourced CosyVoice 3. However, VoxCPM achieves a relatively lower DNSMOS score compared to others, as the prompt audios used in CV3-Hard inherently have a low DNSMOS (around 3.5), which demonstrates its faithful cloning of the recording environment and vibe. These results underscore the model's capability to handle complex, realistic inputs, a strength attributed to the RALM's role in recovering fine-grained acoustic details subsequent to the TSLM-FSQ-based semantic-prosodic modeling.

Subjective evaluations (Table 3) further validate the objective findings, with VoxCPM achieving competitive performance across both languages. On English tests, VoxCPM obtains the highest scores in speaker similarity and good results in naturalness. For Chinese, while VoxCPM trails IndexTTS 2 in naturalness, it achieves slightly superior speaker similarity. We found that many prompt audios from the Seed-TTS-Eval test set contain disfluencies or exhibit a monotonous tone, which inherently constrains the naturalness of our cloned outputs. This pattern suggests that VoxCPM ex-

cels at voice cloning consistency, while IndexTTS 2 may have advantages in prosodic naturalness for Chinese. VoxCPM-Emilia shows competitive speaker similarity but relatively lower naturalness, highlighting the impact of training data scale.

Table 3: Subjective Evaluations in terms of Naturalness and Speaker Similarity. Note: We select the competitive baseline models for subjective comparison based on objective results.

| Model | ZH | | EN | |
|---|---|---|---|---|
| | N-MOS | S-MOS | N-MOS | S-MOS |
| MaskGCT | $3.20 \pm 0.11$ | $3.77 \pm 0.11$ | $3.84 \pm 0.11$ | $4.00 \pm 0.10$ |
| CosyVoice 2 | $3.38 \pm 0.12$ | $4.01 \pm 0.10$ | $4.14 \pm 0.09$ | $3.97 \pm 0.10$ |
| IndexTTS 2 | $4.25 \pm 0.09$ | $4.05 \pm 0.09$ | $4.03 \pm 0.10$ | $4.16 \pm 0.09$ |
| **VoxCPM-Emilia** | $3.79 \pm 0.12$ | $3.99 \pm 0.11$ | $3.91 \pm 0.10$ | $4.10 \pm 0.09$ |
| **VoxCPM** | $4.10 \pm 0.10$ | $4.11 \pm 0.10$ | $4.11 \pm 0.09$ | $4.18 \pm 0.09$ |

### 4.3 ABLATION STUDY: EFFECT OF THE SEMI-DISCRETE BOTTLENECK

As shown in Table 4, the ablation studies on the FSQ bottleneck dimensionality (More details can be found in Appendix Table 9) provide critical insights. The catastrophic performance degradation of the purely continuous model (w/o FSQ), especially on hard cases (ZH-CER: 24.92%), validates our core hypothesis: entangling semantic planning and acoustic rendering in a continuous space leads to instability. Without the inductive bias imposed by FSQ, the model struggles to separate these tasks, resulting in error accumulation on complex utterances.

The optimal performance observed at FSQ levels (FSQ-d128/d256) reveals a key trade-off. Lower dimensions (e.g., FSQ-d4) over-constrain the representation, limiting prosodic capacity. Higher dimensions (e.g., FSQ-d1024) provide insufficient discretization strength, allowing task entanglement to persist. The peak at FSQ-d256 indicates the bottleneck creates an effective "summary space": discrete enough to stabilize long-range semantic planning yet continuous enough to retain crucial prosodic and speaker information, thereby enforcing a beneficial division of labor within the model.

### 4.4 ABLATION STUDY: EFFECT OF RESIDUAL ACOUSTIC MODELING

As shown in Table 4, the ablation studies about the residual language modeling validate our core architectural innovations. Notably, the purely continuous variant (w/o RALM: TSLM $\rightarrow$ LocDiT) —analogous to DiTAR's approach—shows significantly degraded performance, particularly on challenging cases. The performance gap persists across different TSLM configurations, confirming that the challenge is fundamental to the learning objective rather than parameter allocation. This conclusively demonstrates the advantage of our explicit separation between semantic and acoustic modeling. To isolate the benefits of our hierarchical architecture from simple capacity effects, we compared single-stream models with identical initialization foundation but varying depths: [w/o RALM: TSLM(24 layers, LM init.)] vs. [w/o RALM: TSLM(30 layers, partial LM init.)]. The gains from increasing capacity is marginal (e.g., EN-WER from 4.34 to 4.12), while the gains from introducing hierarchical structure is larger in default setting (e.g., EN-WER 2.98). This confirms that the hierarchical TSLM/RALM separation is the dominant factor for stability and expressiveness, significantly outweighing the benefit of merely increased capacity. Furthermore, the hierarchical design exhibits superior performance compared to its single-stream counterpart, even under fully random initialization ([Hierarchical, w/o LM init. in TSLM] vs. [w/o RALM: TSLM(30 layers, random init.)]). This simultaneous improvement demonstrates that the hierarchical residual design provides intrinsic benefit for a specialized division of labor, which is effective even without the strong semantic grounding from pre-trained parameters.

Secondly, the critical role of residual acoustic input is further evidenced by the substantial degradation when ablating original acoustic embeddings (w/o $E_{<i}$ in RALM), highlighting that the RALM requires fine-grained acoustic information to accurately recover acoustic details. Finally, the best performance of the default setting demonstrates the effectiveness of the residual connection. By summing the TSLM and RALM hidden states, the model explicitly delegates semantic-prosodic planning to the TSLM and acoustic refinement to the RALM, achieving optimal integration.

Finally, the effect of pre-training initialization is intuitive: removing pre-trained text LM initialization results in a significant degradation in intelligibility (e.g., EN-WER 2.98 vs. 5.24). This confirms

that the pre-trained knowledge is critical for establishing the TSLM's initial capability and stability. We observe that models trained from random initialization shows a marginal increase in SIM, suggesting that without the strong linguistic guidance from the pre-trained LLM, the model might implicitly allocate more capacity to acoustic feature modeling.

Table 4: Ablation Studies about FSQ bottleneck dimensions and core architecture designs.

| Model Setting | EN | | ZH | | ZH-hard case | |
|---|---|---|---|---|---|---|
| | WER $\downarrow$ | SIM $\uparrow$ | CER $\downarrow$ | SIM $\uparrow$ | CER $\downarrow$ | SIM $\uparrow$ |
| default setting (w/ FSQ: d256s9) | 2.98 | 62.6 | 1.77 | 70.4 | 18.19 | 64.9 |
| w/ FSQ: d4s9 | 5.18 | 59.3 | 4.05 | 68.0 | 19.55 | 62.3 |
| w/ FSQ: d128s9 | 3.43 | 62.2 | 1.67 | 70.7 | 16.76 | 65.7 |
| w/ FSQ: d1024s9 | 3.07 | 62.0 | 2.38 | 69.8 | 20.38 | 64.7 |
| w/o FSQ: d1024s$\infty$ | 3.67 | 62.1 | 2.30 | 69.6 | 24.92 | 63.5 |
| Hierarchical, w/o LM init. in TSLM | 5.24 | 63.4 | 2.41 | 70.9 | 24.66 | 65.6 |
| w/o RALM: TSLM (24 layers, LM init.) $\rightarrow$ LocDiT | 4.34 | 61.8 | 3.05 | 69.4 | 25.00 | 63.8 |
| w/o RALM: TSLM (30 layers, random init.) $\rightarrow$ LocDiT | 5.35 | 62.6 | 3.46 | 69.8 | 30.40 | 63.9 |
| w/o RALM: TSLM (30 layers, partial LM init.)$\rightarrow$ LocDiT | 4.12 | 62.0 | 3.07 | 69.6 | 26.20 | 63.1 |
| w/o $E_{<i}$ in RALM: TSLM $\rightarrow$ ALM $\rightarrow$ LocDiT | 4.91 | 60.9 | 4.94 | 68.1 | 27.17 | 61.7 |
| w/o $h^{\text{residual}}$ in condition: TSLM $\rightarrow$ FSQ $\rightarrow$ LocDiT | 3.86 | 58.3 | 3.05 | 67.6 | 23.65 | 61.7 |

## 4.5 ANALYSIS AND DISCUSSION

**Analysis of Hierarchical Representations** T-SNE visualizations (Appendix Figure 2 to 5) confirm the specialized roles of TSLM and RALM. TSLM-FSQ outputs form semantic-prosodic structures closely tied to text content, whereas RALM residuals exhibit strong speaker-related variations for acoustic rendering. This functional specialization validates the efficacy of our hierarchical residual modeling. More details and analysis can be found in Appendix F.5.

**Expressive Synthesis Capabilities** Beyond quantitative metrics and visualizations, VoxCPM shows good expressive synthesis capabilities directly from text benfiting from the architecture design and training data. When not using prompt speech, the model tends to express suitable style from contextual cues, also shown in F.5. We strongly recommend readers to listen our demo samples.

**Scalability and Efficiency** The performance improvement from VoxCPM-Emilia to VoxCPM highlights the architecture's scalability with increased data. The hierarchical design allows larger models to effectively utilize increased capacity for learning complex patterns. In terms of inference efficiency, VoxCPM-0.5B achieves a real-time factor (RTF) of 0.17 on a single NVIDIA RTX 4090, confirming practical deployment feasibility.

## 5 CONCLUSION

In this work, we resolve the fundamental trade-off between expressivity and stability in text-to-speech synthesis by introducing a unified, end-to-end framework based on hierarchical semantic-acoustic modeling with semi-discrete residual representations. Our approach leverages a differentiable quantization bottleneck to induce a natural separation of concerns: a text-semantic language model captures high-level semantic-prosodic structure, while a residual acoustic model recovers fine-grained details. This eliminates the dependency on external discrete speech tokenizers and mitigates the error accumulation that plagues purely continuous autoregressive models. Extensive experiments demonstrate that our model achieves state-of-the-art zero-shot TTS performance among open-source systems, excelling in both intelligibility and speaker similarity. The success of VoxCPM validates that learning structured, regularized latent spaces provides a principled foundation for expressive generative audio modeling.

## 6 ACKNOWLEDGEMENT

This work is supported by National Natural Science Foundation of China (62076144) and Shenzhen Science and Technology Program (JCYJ20220818101014030).

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

## A  THE USE OF LARGE LANGUAGE MODELS (LLMS)

In accordance with ICLR policy, we disclose that a large language model (LLM), specifically Gemini-2.5 and DeepSeek, was used as a general-purpose assistive tool during the writing of this paper. The LLM was employed solely for the purpose of text polishing and refinement, specifically to assist with improving grammar, enhancing word choice, and increasing the overall readability and fluency of the manuscript. However, the model played no role in the intellectual or scientific contributions of this work, including research ideation, experimental design, result analysis, or conclusion drawing. All such core content is solely the product of the human authors, who take full responsibility for the research presented.

## B  REPRODUCIBILITY STATEMENT

Inference codes are available at codes.zip. Training was conducted using the Megatron framework on 40 NVIDIA H100 GPUs for the main VoxCPM model and 24 H100 GPUs for VoxCPM-Emilia. Ablation studies used 8 H100 GPUs. Full details about hyperparameters are in Section D.

## C  ETHICS STATEMENT

Since our zero-shot TTS model achieves high-quality speech synthesis with the ability to closely mimic speaker characteristics, it carries potential risks of misuse. These risks include, but are not limited to, spoofing voice authentication systems or impersonating a specific speaker without their consent. Our experiments were conducted under the assumption that the use of any reference speaker's voice is authorized and intended for legitimate synthesis purposes. To mitigate these risks, we strongly advocate for the development of robust synthesized speech detection algorithms. Furthermore, we believe it is crucial to establish clear ethical guidelines and reporting mechanisms for the responsible deployment of such technology.

## D  IMPLEMATATION DETAILS OF VOXCPM

### D.1  MODEL ARCHITECTURE

VoxCPM consists of a 24-layer TSLM (initialized from MiniCPM-4-0.5B) and a 6-layer RALM. The FSQ layer uses 256 dimensions with 9 scalar levels. The diffusion decoder has 4 layers, optimized for high-efficacy latent generation, as shown in Table 5

For Audio VAE, it operates at a 25 Hz frame rate, designed to be compatible with the streaming nature of VoxCPM. The VAE's architecture is inspired by DAC, with both its encoder and decoder implemented using stacked Causal Convolutional Networks (Causal CNNs). For 16 kHz single-channel audio, the encoder achieves a 640x downsampling factor through a series of strided convolutions with a stride sequence of [2, 5, 8, 8], compressing the audio into a 25 Hz latent representation. The decoder then reconstructs the original waveform by upsampling from this latent representation. The training objectives consist of an adversarial (GAN) loss, a Mel-spectrogram loss, and a KL divergence loss, with the latter's weight set to 5e-5.

### D.2  TRAINING CONFIGURATION

Both VoxCPM and VoxCPM-Emilia employed the Warmup-Stable-Decay (WSD) training strategy, which we found essential for optimal convergence. Specifically, the decay phase with annealing to a very low learning rate (combined with batch size doubling) significantly enhances model performance, particularly for zero-shot speaker similarity, as demonstrated in Table 10. All ablation studies followed the same 200K-step training protocol on 8 H100 GPUs using the Emilia dataset, employing a fixed learning rate of $1 \times 10^{-4}$ to ensure consistent comparisons.

Table 5: The model architecture and parameters of VoxCPM-0.5B.

| Module | Configuration | Parameters |
|---|---|---|
| LocEnc | 4 layers, 1024 hidden dim, 4096 FFN dim | 59M |
| TSLM | 24 layers, 1024 hidden dim, 4096 FFN dim | 433M |
| FSQ | 256 dimensions, 9 quantization levels | 0.5M |
| RALM | 6 layers, 1024 hidden dim, 4096 FFN dim | 89M |
| LocDiT | 4 layers, 1024 hidden dim, 4096 FFN dim | 64M |
| Stop Predictor | 3-layer MLP, 1024 hidden dim, 2 output dim | 1M |
| patch-size | 2 (that is, TSLM and RALM work in 12.5Hz token rate) | - |
| AudioVAE | 16kHz waveform $\rightarrow$ 25Hz latents (downsampling at [2, 5, 8, 8]) | 75M |

Table 6: Training configurations for VoxCPM variants.

| Model | Phase | Learning Rate | Tokens/Batch | Steps | GPUs |
|---|---|---|---|---|---|
| VoxCPM | Stable | $1 \times 10^{-4}$ | 4,096 | 400K | $40 \times$ H100 |
| VoxCPM | Decay | $1 \times 10^{-4} \rightarrow 5 \times 10^{-6}$ | 8,192 | 100K | $40 \times$ H100 |
| VoxCPM-Emilia | Stable | $1 \times 10^{-4}$ | 4,096 | 150K | $24 \times$ H100 |
| VoxCPM-Emilia | Decay | $1 \times 10^{-4} \rightarrow 5 \times 10^{-6}$ | 8,192 | 50K | $24 \times$ H100 |
| VoxCPM-ablation | Stable | $1 \times 10^{-4}$ | 4,096 | 200K | $8 \times$ H100 |

## E   DETAILS OF BASELINES

We compared VoxCPM against several state-of-the-art TTS systems, focusing on open-source models from the SEED-TTS-EVAL and CV3-EVAL benchmarks. Below, we describe the key baselines, their architectures, training data, and how evaluation samples were generated.

- **CosyVoice Series** (Du et al., 2024a;b; 2025): A family of TTS models leveraging supervised semantic tokens and multi-stage pipelines. CosyVoice (Du et al., 2024a) uses a transformer-based architecture with S3Tokenizer, trained on approximately 170K hours of multilingual speech data (primarily English and Chinese). CosyVoice2 (Du et al., 2024b) enhances this with text-based LLM initialization, and introduces streaming synthesis for low-latency synthesis. CosyVoice3 (Du et al., 2025) further leverages in-the-wild speech data, and proposes a novel speech tokenizer to capture prosody, with powerful post-training techniques. We used officially released checkpoints from `https://github.com/FunAudioLLM/CosyVoice` with default settings to generate samples.

- **MaskGCT** (Wang et al., 2025c): A non-autoregressive transformer model that employs masked generative transformers to predict discrete speech tokens derived from a neural audio codec and SSL tokens. It is trained on the Emilia dataset and achieves a strong performance on zero-shot TTS. Besides, it can control the duration precisely. Samples were generated using the official implementation provided at `https://github.com/open-mmlab/Amphion/tree/main/models/tts/maskgct`.

- **F5-TTS** (Chen et al., 2024): A non-autoregressive model utilizing flow-matching for efficient speech synthesis. It operates on continuous mel-spectrogram representations, also trained on Emilia dataset. We used the official codes and checkpoint from `https://github.com/SWivid/F5-TTS` to generate evaluation samples.

- **SparkTTS** (Wang et al., 2025b): A languae model-based model using single-stream decoupled speech tokens to improve modeling efficiency. It is trained on 100K hours of bilingual (English and Chinese) dataset VoxBox, enabling it capability to achieve controllable speech synthesis. Samples were generated using the official implementation at `https://github.com/SparkAudio/Spark-TTS`.

- **FireRedTTS Series** (Guo et al., 2024; Xie et al., 2025): FireRedTTS presents a foundation TTS framework for industry-level generative speech application. FireRedTTS2 fur-

ther employs it for long-form multi-speaker dialogue. FireRedTTS-2 improves scalability with a refined transformer architecture, trained on 1.1M hours of monologue speech data and 300k hours of multi-speaker dialogue data. We used official checkpoints from `https://github.com/FireRedTeam/FireRedTTS` for sample generation.

- **IndexTTS2** (Zhou et al., 2025): An autoregressive large-scale TTS model with precise duration control. It also achieves disentanglement between emotional expression and speaker identity, enabling independent control over timbre and emotion. Samples were generated using the official checkpoint from `https://github.com/index-tts/index-tts`.

- **Higgs Audio v2**: A powerful audio foundation model pretrained on over 10 million hours of audio data and a diverse set of text data. It proposes a unified audio tokenizer captures both semantic and acoustic features. We used the official implementation at `https://github.com/boson-ai/higgs-audio` to generate samples.

- **VibeVoice** (Peng et al., 2025):A novel framework designed for generating expressive, long-form, multi-speaker conversational audio, such as podcasts, from text. It presents a continuous speech tokenizers (Acoustic and Semantic) operating at an ultra-low frame rate of 7.5 Hz. Because now the codes are unavailable at `https://github.com/microsoft/VibeVoice`, we use the official results reported in the paper.

- **OpenAudio-s1-mini**: A compact and powerful TTS model (0.5B parameters) using dual-AR architecture and online Reinforcement Learning from Human Feedback (RLHF). Samples were generated using the official implementation at `https://github.com/fishaudio/fish-speech`.

# F ADDITIONAL EXPERIMENTAL RESULTS

## F.1 COMPREHENSIVE COMPARISONS ON SEED-TTS-EVAL BENCHMARK

In this section, we present comprehensive comparisons covering both open-source and close-source TTS systems. As shown in Table 7, VoxCPM not only outperforms all open-source competitors but also achieves results comparable to several proprietary systems, despite their significantly larger parameter counts and training data scales.

Table 7: Performance on Seed-TTS-eval Benchmark, including close-sourced systems

| Model | Params | Open-Source | EN | | ZH | | Hard | |
|---|---|---|---|---|---|---|---|---|
| | | | WER ↓ | SIM ↑ | CER ↓ | SIM ↑ | CER ↓ | SIM ↑ |
| MegaTTS3 | 0.5B | ✗ | 2.79 | 77.1 | 1.52 | 79.0 | - | - |
| DiTAR | 0.6B | ✗ | 1.69 | 73.5 | 1.02 | 75.3 | - | - |
| CosyVoice3 | 0.5B | ✗ | 2.02 | 71.8 | 1.16 | 78.0 | 6.08 | 75.8 |
| CosyVoice3 | 1.5B | ✗ | 2.22 | 72.0 | 1.12 | 78.1 | 5.83 | 75.8 |
| Seed-TTS | - | ✗ | 2.25 | 76.2 | 1.12 | 79.6 | 7.59 | 77.6 |
| MiniMax-Speech-02 | - | ✗ | 1.65 | 69.2 | 0.83 | 78.3 | - | - |
| F5-TTS | 0.3B | ✓ | 2.00 | 67.0 | 1.53 | 76.0 | 8.67 | 71.3 |
| MaskGCT | 1B | ✓ | 2.62 | 71.7 | 2.27 | **77.4** | - | - |
| CosyVoice | 0.3B | ✓ | 4.29 | 60.9 | 3.63 | 72.3 | 11.75 | 70.9 |
| CosyVoice2 | 0.5B | ✓ | 3.09 | 65.9 | 1.38 | 75.7 | **6.83** | 72.4 |
| SparkTTS | 0.5B | ✓ | 3.14 | 57.3 | 1.54 | 66.0 | - | - |
| FireRedTTS | 0.5B | ✓ | 3.82 | 46.0 | 1.51 | 63.5 | 17.45 | 62.1 |
| FireRedTTS-2 | | ✓ | 1.95 | 66.5 | 1.14 | 73.6 | - | - |
| Qwen2.5-Omni | 7B | ✓ | 2.72 | 63.2 | 1.70 | 75.2 | 7.97 | 74.7 |
| OpenAudio-s1-mini | 0.5B | ✓ | 1.94 | 55.0 | 1.18 | 68.5 | 23.37 | 64.3 |
| IndexTTS 2 | 1.5B | ✓ | 2.23 | 70.6 | 1.03 | 76.5 | 7.12 | **75.5** |
| VibeVoice | 1.5B | ✓ | 3.04 | 68.9 | 1.16 | 74.4 | - | - |
| HiggsAudio-v2 | 3B | ✓ | 2.44 | 67.7 | 1.50 | 74.0 | 55.07 | 65.6 |
| **VoxCPM-Emilia** | 0.5B | ✓ | 2.34 | 68.1 | 1.11 | 74.0 | 12.46 | 69.8 |
| **VoxCPM** | 0.5B | ✓ | **1.85** | **72.9** | **0.93** | 77.2 | 8.87 | 73.0 |

## F.2 Effect of LM Guidance on LocDiT

To explore CFG influence and select the best inference setting, we tested different CFG value, that is, the LM (the sum of TSLM-FSQ hidden and RALM hidden) guidance on LocDiT. We found that CFG value=2.0 could achieve optimal balance across all metrics. Higher weights ($\geq$3.0) degraded intelligibility significantly.

Table 8: Effect of LM guidance on LocDiT, tested on Seed-TTS-eval Benchmark.

| CFG Value | EN | | ZH | | ZH-hard case | |
|---|---|---|---|---|---|---|
| | WER ↓ | SIM ↑ | CER ↓ | SIM ↑ | CER ↓ | SIM ↑ |
| 1.0 (w/o CFG) | 16.32 | 55.1 | 14.47 | 61.5 | 56.87 | 43.0 |
| 1.5 | 1.86 | 72.1 | 1.16 | 77.0 | 9.60 | 73.9 |
| 2.0 | 1.85 | 72.9 | 0.93 | 77.2 | 8.87 | 73.0 |
| 3.0 | 2.16 | 71.4 | 1.12 | 74.7 | 13.22 | 65.0 |
| 5.0 | 12.78 | 60.7 | 17.23 | 59.4 | 48.46 | 39.9 |

## F.3 Effect of the semi-discrete bottleneck

This section shows more comprehensive investigation about FSQ bottleneck dimension, as shown in Table 9. It can be found that when removing scalar quantization in the bottleneck layer, the performance tends to degrade, showing that semi-discrete latent space maybe more stable and robust than fully continuous space. Besides, we investigated the impact between FSQ and VAE regularization as the bottleneck in VoxCPM. We replaced the FSQ bottleneck with a continuous VAE bottleneck and trained it under the same setting. We specifically chose a dimension of 16 for the continuous bottleneck (16d-VAE), as high-dimensional continuous latents (e.g., 256d) empirically suffer from posterior collapse or inefficient compression due to optimization challenges. Under the same 16-dimensional setting, the FSQ model achieves significantly better robustness than the VAE. The VAE bottleneck nearly doubles the error rate on challenging cases (CER=28.17%) compared to the FSQ counterpart (CER=14.42%). This confirms that the semi-discrete nature of FSQ provides an indispensable inductive bias for stability that continuous regularization lacks. While the continuous 16d-VAE achieves slightly higher English speaker similarity than 16d-FSQ (likely because continuous vectors can encode finer acoustic details), this comes at the cost of intelligibility. Our default 256d-FSQ strikes the best balance, maintaining high expressivity without the optimization instability associated with high-dimensional continuous bottlenecks.

Table 9: FSQ dimension selection study on the Emilia dataset. *Note:* The 256-dim configuration was selected for the final VoxCPM configuration with the understanding that larger training datasets needs more powerful modeling capabilities.

| Model Setting | EN | | ZH | | ZH-hard case | |
|---|---|---|---|---|---|---|
| | WER ↓ | SIM ↑ | CER ↓ | SIM ↑ | CER ↓ | SIM ↑ |
| w FSQ: d4s9 | 5.18 | 59.3 | 4.05 | 68.0 | 19.55 | 62.3 |
| w FSQ: d16s9 | 3.22 | 60.4 | 1.87 | 70.5 | **14.42** | **66.2** |
| w FSQ: d64s9 | 3.22 | 61.1 | 2.14 | 69.8 | 17.48 | 65.1 |
| w FSQ: d128s9 | 3.43 | 62.2 | **1.67** | **70.7** | 16.76 | 65.7 |
| w FSQ: d256s9 | **2.98** | **62.6** | 1.77 | 70.4 | 18.19 | 64.9 |
| w FSQ: d1024s9 | 3.07 | 62.0 | 2.38 | 69.8 | 20.38 | 64.7 |
| w/o FSQ: d1024s∞ | 3.67 | 62.1 | 2.30 | 69.6 | 24.92 | 63.5 |
| w VAE: d16 | 3.56 | 62.1 | 1.94 | 69.7 | 28.17 | 62.7 |

## F.4 Effect of Training Phase on Performance

As mentioned in D.2, the two-phase Warmup-Stable-Decay (WSD) learning rate schedule is critical for achieving optimal model performance. The initial Stable phase allows the model to converge reliably to a strong baseline. The subsequent Decay phase is then essential for refining the model,

particularly for improving its zero-shot voice similarity capabilities. Furthermore, doubling the batch size (from 4K to 8K tokens) during the Decay phase is a necessary complement to the reduced learning rate. A larger batch size provides a more accurate and stable estimate of the gradient direction, which is crucial for effective and stable optimization when using very low learning rates. This strategy prevents the noise from small-batch gradients from destabilizing the fine-tuning process, enabling the model to make consistent improvements in both intelligibility (lower WER) and speaker similarity (higher SIM).

Table 10: Performance across training phases.

| Phase | EN | | ZH | | ZH-Hard Case | |
|---|---|---|---|---|---|---|
| | WER $\downarrow$ | SIM $\uparrow$ | CER $\downarrow$ | SIM $\uparrow$ | CER $\downarrow$ | SIM $\uparrow$ |
| Stable | 2.05 | 69.7 | 0.99 | 75.1 | 13.22 | 68.6 |
| Decay | 1.85 | 72.9 | 0.93 | 77.2 | 8.87 | 73.0 |

### F.5 VISUAL ANALYSIS OF HIERARCHICAL REPRESENTATIONS

To validate our core hypothesis of learned implicit semantic-acoustic disentanglement, we conducted a t-SNE visualization of the internal representations in our hierarchical model. The resulting distributions, shown in Figures 2 and 3, empirically confirm the specialized roles of the TSLM and the RALM. Figure 2 illustrates the model's behavior in a zero-shot voice cloning task, where each color corresponds to a distinct utterance from an unseen speaker. The TSLM-FSQ outputs form a stable, speaker-agnostic semantic-prosodic structure, while the RALM residuals cluster by speaker identity, confirming their specialized roles in content planning and acoustic refinement.

Figure 3 further demonstrates the VoxCPM's capability to infer appropriate prosody and style directly from text, when not using any speech prompt. When processing different text genres (news, poetry, conversation), TSLM-FSQ representations cluster by semantic category, showing that the pre-trained language model backbone effectively infers appropriate prosodic patterns directly from text content. For example, embeddings for "news" group together, separate from "story-telling" or "rap-lyrics." The RALM outputs display greater within-category variation, indicating its role in adding fine-grained acoustic nuances to the semantic-prosodic plan.

To investigate whether the observed semantic-acoustic specialization originates from the hierarchical structure design or the pre-trained LLM weights, we conducted a controlled experiment. We trained a VoxCPM variant with fully random TSLM initialization and performed t-SNE visualization of its latent spaces for both zero-shot cloning (Figure 4) and text-to-speech (Figure 5) tasks.

The visualizations reveal that even in the absence of pre-trained semantic knowledge, the latent space exhibits a similar functional division of labor. The TSLM-FSQ hidden states still show relevance to semantic content and weak speaker relevance (as seen in Figure 5), while the RALM hidden states display strong speaker-dependent clustering (as seen in Figure 4). Although this self-learned specialization is slightly less distinct compared to the LLM initialized model, the consistency confirms that the residual semi-discrete bottleneck structure itself is the key inductive bias that enables the model to self-adapt and learn a specific semantic-acoustic division of labor. The pre-trained language model parameters primarily serve to significantly sharpen, stabilize, and enhance this inherent modeling capacity.

### F.6 FUSION STRATEGIES ABLATION

The LocDiT module is responsible for generating the final speech latents by conditioning on the combined semantic-acoustic representation derived from the TSLM (via FSQ) and the RALM. In the default VoxCPM configuration, we adopt a simple element-wise summation of the TSLM and RALM output hidden states as the fusion mechanism, primarily for its intuitively aligning with the "residual" nature of the design and good performance. To validate this choice, we conducted an ablation study comparing summation against several other common feature fusion strategies (concatenation, gating, and attention-based fusion). The results are presented in Table 11. It shows that while **concatenation** slightly improves speaker similarity, the simple **summation** strategy achieves

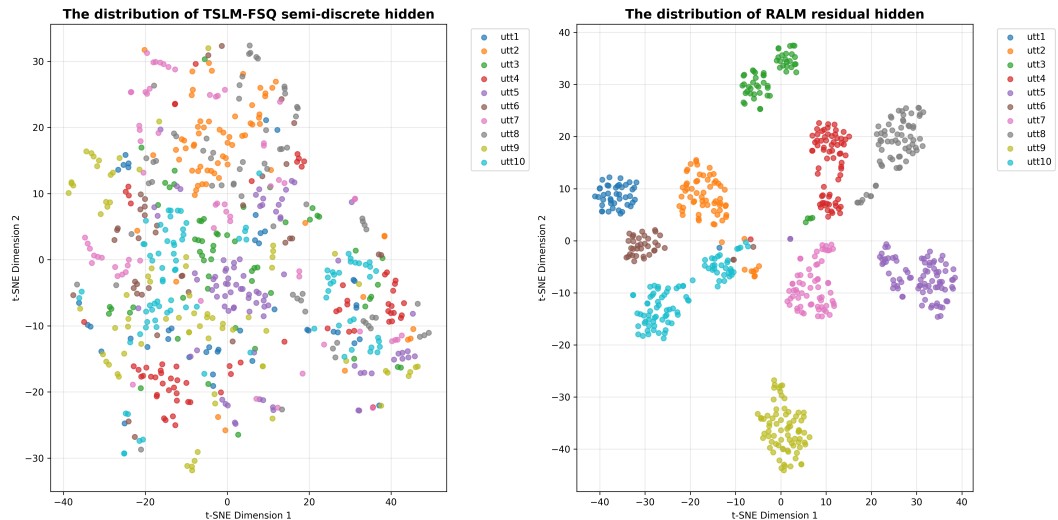

Figure 2: The T-SNE visualization of latent space distributions in zero-shot voice cloning task.

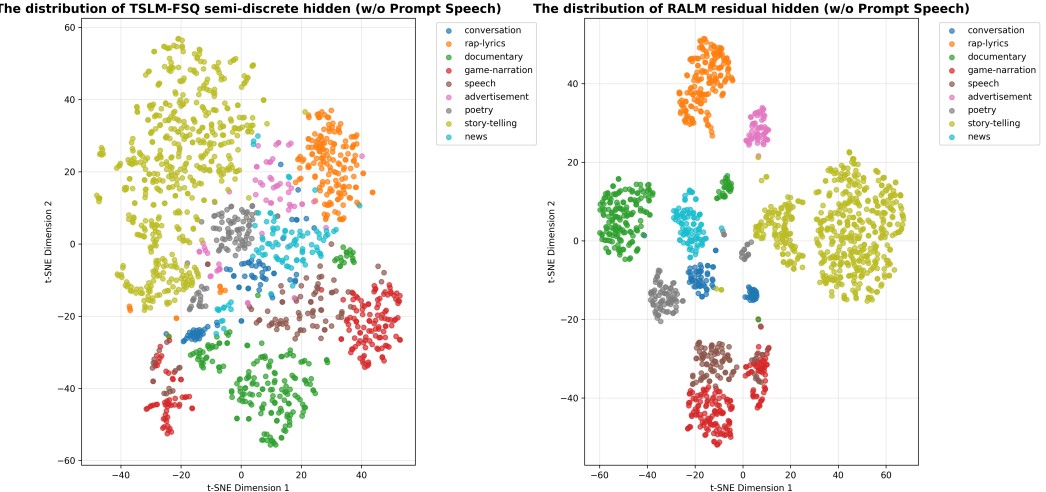

Figure 3: The T-SNE visualization of latent space distributions in text-to-speech task, without prompt speech.

the best balance of intelligibility (lowest WER and CER) and acoustic quality. The significantly higher error rates observed with attention-based fusion suggest that relying on the model to dynamically learn the fusion weights introduces optimization instability in the continuous latent space.

Table 11: Ablation of different fusion strategies for combining TSLM and RALM outputs in LocDiT.

| Fusion Strategy | EN-WER ($\downarrow$) | EN-SIM ($\uparrow$) | ZH-CER ($\downarrow$) | ZH-SIM ($\uparrow$) |
|---|---|---|---|---|
| **Sum (Default)** | **2.98** | 62.6 | **1.77** | 70.4 |
| Concatenation | 3.16 | **64.1** | 2.88 | **71.3** |
| Gating | 3.75 | 63.6 | 2.04 | 70.2 |
| Attention | 4.92 | 59.9 | 5.43 | 66.5 |

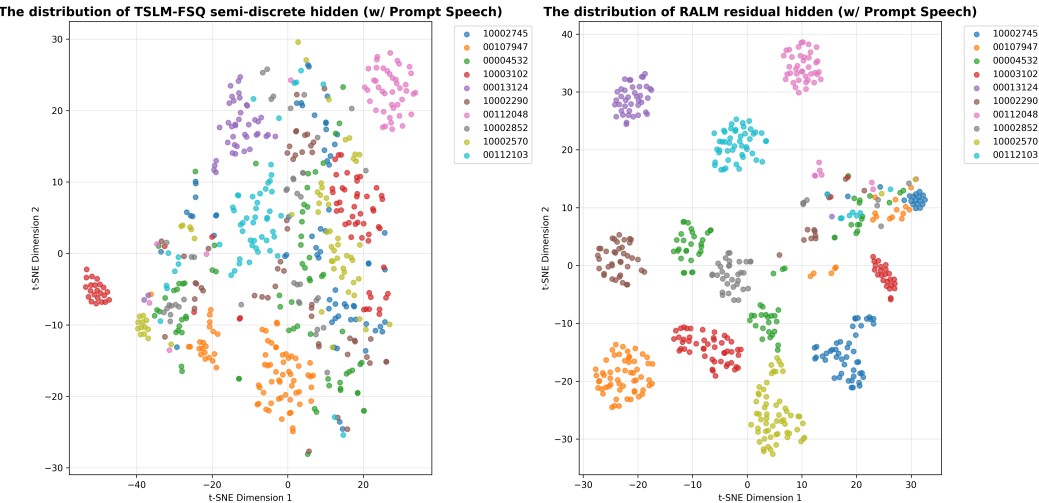

Figure 4: The T-SNE visualization of latent space distributions (without pretrained LLM parameters initialization for TSLM) in zero-shot voice cloning task.

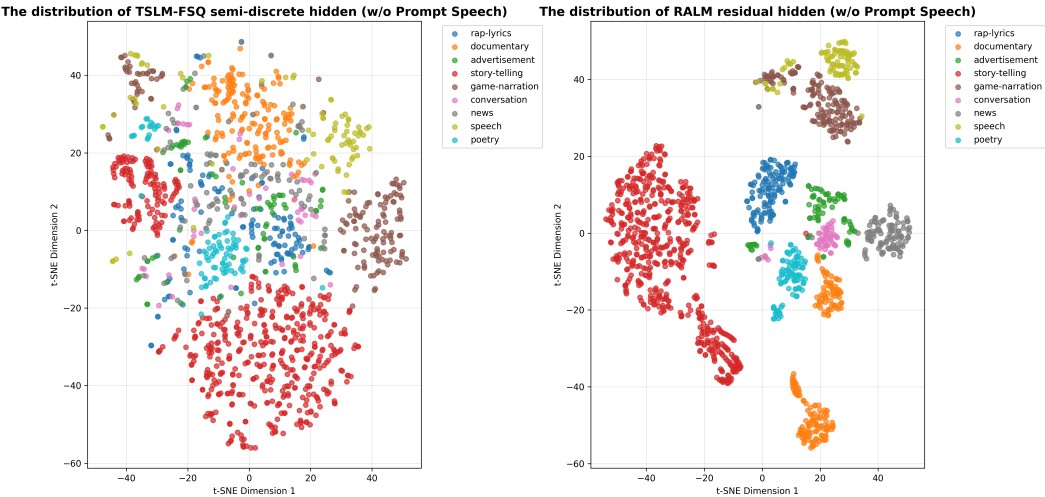

Figure 5: The T-SNE visualization of latent space distributions (without pretrained LLM parameters initialization for TSLM) in text-to-speech task, without prompt speech.

## F.7    PROBING ANALYSIS FOR SEMANTIC-ACOUSTIC MODELING

To verify the core hypothesis that the FSQ bottleneck induces a functional separation of labor between the TSLM (semantic focus) and RALM (acoustic focus), we conducted **Layer-wise Probing Experiments** following the SUPERB (Yang et al., 2021) setting. We trained lightweight linear classifiers on the frozen internal hidden states extracted from four key locations: LocEnc, TSLM , FSQ, and RALM. We assessed the capture of linguistic-content using phoneme recognition (PR) and automatic speech recognition (ASR), measured by Phoneme Error Rate (PER) and WER, respectively. The lower values indicate better linguistic content preservation (higher intelligibility). Acoustic-timbre modeling was evaluated via the Automatic Speaker Verification (ASV) task, measured by Equal Error Rate (EER). The lower values indicate better speaker identity modeling (higher acoustic/timbre distinguishability).

The probing experiment results offer evidence to our core architectural claims:

- Acoustic Features Aggregation: The LocEnc is responsible for aggregating the low-level continuous acoustic latents from AudioVAE. Consequently, it reflects the weak semantic relevance (PER=59.12%, WER=65.79%) and carries fundamental acoustic information like timbre (EER=15.38%).

- Semantic Specialization: The TSLM representation achieves the lowest PER (45.60%) and WER (60.43%), validating its role in capturing the stable linguistic and prosodic content.

- Acoustic Filtering: The FSQ representation exhibits the highest EER (19.25%). This is a critical finding: the quantization bottleneck effectively acts as an acoustic filter, actively reducing correlation with speaker identity and prioritizing content-prosodic stability.

- Residual Recovery: Conversely, the RALM representation achieves the lowest EER (13.24%). This conclusively proves that the RALM specializes in recovering the fine-grained acoustic residuals and speaker identity that were filtered by the FSQ.

The slight rise in content error post-FSQ (PER=50.90%, WER=62.37%) confirms the expected lossy nature of quantization and the subsequent shift in the RALM's focus from pure semantic modeling toward acoustic refinement. These findings empirically validate our premise: VoxCPM successfully enforces a natural division of labor using the FSQ as an inductive bias.

Table 12: Layer-wise probing results on internal hidden states of VoxCPM.

| Hidden State Location | PR (PER) ↓ | ASR (WER) ↓ | ASV (EER) ↓ |
|---|---|---|---|
| LocEnc output | 59.12 | 65.79 | 15.38 |
| TSLM last hidden (Pre-FSQ) | **45.60** | **60.43** | 18.70 |
| FSQ output | 50.90 | 62.37 | 19.25 |
| RALM last hidden | 53.49 | 64.85 | **13.24** |

## F.8 EFFECT OF MODEL SCALABILITY

We conducted a model scalability analysis to empirically demonstrate the robustness of the hierarchical VoxCPM architecture across varying capacities. We trained larger variants of VoxCPM, namely 1B and 3B parameters, on the Emilia dataset for 200k iterations.

The specific architectural configurations used for the larger models were as follows:

- **VoxCPM-1B**: This model consisted of a 28-layer TSLM ($h = 2048$), a 7-layer RALM ($h = 2048$), and 6 layers for both the LocEnc and LocDiT modules ($h = 1024$).

- **VoxCPM-3B**: This model was configured with a 32-layer TSLM ($h = 2560$), an 8-layer RALM ($h = 2560$), and 8 layers for both LocEnc and LocDiT ($h = 1024$).

As shown in Table 13, increasing the model capacity consistently improved performance across all core metrics (EN-WER, EN-SIM, ZH-CER, and ZH-SIM). Specifically, the 3B variant achieved the best intelligibility (2.60% WER) and speaker similarity (66.7% SIM). This validates that the multi-component, hierarchical design of VoxCPM remains fully scalable and effectively leverages increased parameter counts.

Table 13: Scalability analysis of VoxCPM variants trained on the Emilia dataset.

| Model Size | EN | | ZH | |
|---|---|---|---|---|
| | WER ↓ | SIM ↑ | CER ↓ | SIM ↑ |
| 0.5B | 2.98 | 62.6 | 1.77 | 70.4 |
| 1B | 2.95 | 65.9 | 1.82 | 72.0 |
| 3B | 2.60 | 66.7 | 1.78 | 72.3 |

## F.9 The Comparison of Real-Time Factor and Streaming Synthesis Details

We evaluated the Real-Time Factor (RTF) of VoxCPM in comparison to other TTS models on a single NVIDIA RTX 4090 GPU. VoxCPM achieves a remarkably low RTF of 0.17, significantly faster than CosyVoice2 (0.52) and SparkTTS (0.80), due to its efficient 12.5 Hz token rate and lightweight components. The throughput for the LM backbone in VoxCPM is approximately 73 tokens/s.

Table 14: The Real-Time Factor of some TTS systems.

| TTS System | RTF $\downarrow$ |
|---|---|
| CosyVoice 2 | 0.52 |
| SparkTTS | 0.80 |
| IndexTTS 2 | 0.85 |
| VoxCPM | **0.17** |

Furthermore, VoxCPM supports true streaming synthesis with a theoretical first-packet latency below 100 ms: the LocDiT generates patches in under 10ms with 10 iterations, owing to its short local context and lightweight parameters, while the causal AudioVAE enables incremental processing. To ensure smooth playback, the last three latents are buffered, resulting in 80 ms audio chunks per step.

## G Further Discussion

### G.1 Theoretical grounding: information bottleneck principles

To better explain the success use of FSQ bottleneck in VoxCPM, we provide an intuitive theoretical grounding information bottleneck principles: The FSQ layer functions as a capacity-constrained bottleneck between the high-capacity TSLM and the downstream RALM. Mathematically, the scalar quantization inherent in FSQ acts as a lossy compression filter. To minimize the end-to-end diffusion loss, the TSLM is heavily penalized if it attempts to encode high-variance, fine-grained acoustic details (like timbre or specific residuals), as these would be significantly distorted by the scalar quantization step. Consequently, the TSLM is naturally forced to prioritize encoding low-variance, stable features—namely, the semantic and global prosodic structure (content, intonation). This mechanism effectively offloads the task of modeling high-frequency, complex acoustic residuals to the RALM.

Empirical results from our probing experiments (Appendix F.7) validate this induced division of labor: TSLM representations demonstrate relatively high semantic accuracy (low PER and WER) but are acoustic-agnostic (high speaker EER), while RALM exhibits the opposite behavior. This confirms the FSQ's effectiveness as an internal inductive bias for hierarchical task separation.

### G.2 Analysis on Quantization Ceiling

**Continuous Latents vs. Discrete Codebooks**: Discrete tokenizers like S3Tokenizer map speech signals to a fixed, finite codebook of size K, imposing a rigid, finite ceiling on expressive information (the "quantization ceiling"). Our AudioVAE operates in a 64-dimensional continuous latent space, which retains significantly richer acoustic detail and avoids the critical precision loss inherent in mapping to a finite, low-dimensional set.

**Internal FSQ Bottleneck Capacity**: Inside VoxCPM, the FSQ bottleneck serves as a regularizer, not a prediction target for a small codebook. We use a high-dimensional FSQ (e.g., d256s9), offering a vast representational space far exceeding traditional VQ/FSQ codebooks. Our FSQ ablation study (Table 9) shows that when we constrain the FSQ capacity to mimic a traditional small codebook (e.g., FSQ d4s9, equivalent to 6561 vocab), the clone similarity (SIM) significantly degrades, proving that full information preservation is essential for the model's performance ceiling.

**VAE Reconstruction Quality**: To quantify the potential information bottleneck imposed by the VAE, we evaluated its reconstruction quality on the SEED-TTS-Eval dataset. The VAE is designed to be reconstruction-oriented with a low KL-penalty, ensuring maximum information retention. As shown in Table 15, the VAE-reconstructed audio exhibits high fidelity across objective metrics. The minor difference between the Ground Truth (GT) and Reconstruction confirms that the continuous

latent space effectively preserves necessary acoustic detail, distinguishing it from the significant information loss typically associated with low-bitrate discrete tokenizers.

Table 15: AudioVAE reconstruction performance comparison against Ground Truth (GT) on SEED-TTS-Eval.

| Metric | Mel Loss ($\downarrow$) | STOI ($\uparrow$) | WER ($\downarrow$) | SIM ($\uparrow$) | DNSMOS ($\uparrow$) |
|---|---|---|---|---|---|
| GT | 0.000 | 1.000 | 1.21 | 75.5 | 3.96 |
| Recon | 0.924 | 0.948 | 1.33 | 73.6 | 3.92 |

## H EVALUATION METRICS AND QUESTIONNAIRES

To ensure robust subjective evaluation, we developed an automated interface for assessing generated speech samples. For each evaluation item, participants interact with three components: the Evaluation Interface (containing the audio sample to be rated), the Questionnaire, and the Rating Guidelines. Evaluations were conducted by 20 people for both English and Chinese samples.

### H.1 NATURALNESS MOS

**Evaluation Interface**: A single audio clip accompanied by its corresponding text prompt.

**Questionnaire**: To what extent does the speech sound natural and human-like? Does it convey engagement with the content, or does it resemble an artificial voice lacking contextual understanding?

**Rating Guidelines**:

- 5: Indistinguishable from human speech, highly natural and engaging.
- 4: Surpasses expectations for synthetic speech, very human-like.
- 3: Meets expectations for AI-generated speech, reasonably natural.
- 2: Below average, with noticeable artificial qualities.
- 1: Clearly synthetic, lacking human-like expression.

### H.2 SPEAKER SIMILARITY MOS

**Evaluation Interface**: A reference audio clip alongside the audio clip to be evaluated.

**Questionnaire**: Focusing solely on voice characteristics (disregarding content and audio quality), how closely does the evaluated voice match the reference voice?

**Rating Guidelines**:

- 5: Nearly identical to the reference voice, as if spoken by the same person.
- 4: Highly similar to the reference voice, with minor differences.
- 3: Moderately similar, sharing some vocal traits.
- 2: Largely dissimilar, with few shared characteristics.
- 1: Completely distinct, bearing no resemblance to the reference voice.

