# OpenReview forum: "Hierarchical Semantic-Acoustic Modeling via Semi-Discrete Residual Representations for Expressive End-to-End Speech Synthesis"
_ICLR.cc/2026/Conference — ICLR 2026 Poster_

### Official Review · Reviewer_MsWh · 2025-10-20

**Soundness:** 3
**Presentation:** 3
**Contribution:** 2
**Rating:** 4
**Confidence:** 4

**Summary:**

The paper proposes a cascaded TTS model that predicts semantic tokens, refine with added residual latents and finally decoded into speech latents with a latent diffusion model.

**Strengths:**

1. While there are many similar cascaded architecture that predicts semantic tokens, refine with acoustic tokens/latents and finally decode into speech latents/waveform, there is some (but limited) novelty in using FSQ as a bottleneck layer directly after LLM output, rather than as in a pre-trained speech tokenizer. This does give some stability and flexibility.
2. There is ablation study that investigate the impacts of the design choices.

**Weaknesses:**

1. While the model does seem to perform better in term of WER, the speech naturalness performance seem to be a mixed bag. In Table 2, the model does not perform as well in DNSMOS metrics. Also in Table 3 the model underperforms/perform on-par on naturalness. In the evaluation section the authors tends to spent more time on the positive results but discuss little about the negative results. I also want to ask why some of the models evaluated in Table 2 are not presented in Table 3?
2. The paper doesn't evaluate the expressiveness nor the controllability of the proposed model. Some competitors (e.g. CozyVoice and Higgs Audio) are focusing on these aspects, and it's an important direction in the field of TTS. I wonder if the authors have some results in this area?

**Questions:**

See weakness

---

> ### Author Response · Authors · 2025-11-24
> **Response to Reviewer MsWh**
>
> Thank you for your constructive feedback and for recognizing the novelty in using FSQ and the importance of ablation study. We appreciate the opportunity to clarify our methodological contributions and the nagative results. We hope the following responses fully resolve your concerns.
>
> > **Novelty Concerns about Cascaded vs. E2E**
>
> We respectfully clarify that VoxCPM is not a standard cascaded system. Unlike cascaded systems (e.g., CosyVoice: LM+SpeechTokenizer+Diffusion) where modules are trained disjointly, VoxCPM is an **end-to-end differentiable framework**. Gradients from the final diffusion loss propagate back through the LocDiT, RALM, FSQ,TSLM and LocEnc, allowing the semantic planner to adapt specifically for the acoustic renderer. This design achieves a level of task coordination that disjoint cascaded systems cannot match.
>
> Other reviewers have affirmed the innovation of this approach, describing it as an "elegant and well-motivated solution" to task entanglement (ada6), a "conceptually coherent and technically sound" framework (emjo), one that "overcomes the limits of both discrete and continuous methods" (kJR8), and a "natural evolution" with "cohesive hierarchical design" and "innovative FSQ usage" (Nvjv). We sincerely hope you might reconsider your assessment regarding the novelty of our method.
>
> > **W1/Q1: Analysis of Negative Results**
>
> We appreciate your attention for raising this. It is important to distinguish **"Faithful Cloning"** from **"Audio Enhancement."**
>
> Upon re-examination, we found that the *prompt audios used in CV3-Hard have a relatively low DNSMOS (3.52 for Chinese, 3.55 for English)*. VoxCPM achieves a similar DNSMOS result, indicating faithful cloning of the recording environment/vibe.
> Similarly, in the Seed-TTS-Eval test set, *many prompt audios contain disfluencies or exhibit a monotonous tone*, which inherently constrains the naturalness of our cloned outputs. As a result, VoxCPM does not achieve the highest N-MOS ratings in subjective evaluations—a direct outcome of our high-fidelity cloning objective.
>
> While some competitors achieve higher DNSMOS and naturalness results, this often alters speaker identity (as seen in lower SIM or S-MOS). In contrast, we prioritize the preservation of the original speaker’s acoustic characteristics, including potential noise and disfluencies, over artificially "cleaning" the signal, in order to achieve more authentic zero-shot voice cloning.
>
> Thank you for your insightful comment, which prompted us to conduct a detailed re-analysis of the objective and subjective naturalness metrics. We have **integrated this detailed analysis into Section 4.2** of the revised version.
>
> > **W1/Q1: Selection of Models for Subjective Evaluation**
>
> Due to high costs of rigorous human MOS testing, we could not include every baseline from the objective benchmarks. We selected the **top-performing open-source systems** (CosyVoice2, IndexTTS2, MaskGCT) to ensure a fair comparison against the strongest available competitors. We have **added a note in the caption of Table 3 to clarify this selection criterion**.
>
> > **W2/Q2: Expressiveness and Controllability.**
>
> Thank you for raising the importance of controllability. While our primary focus is on establishing the **foundational architecture for TTS**, we agree that controllability is a vital direction for future research.
>
> We would like to clarify VoxCPM is designed as a foundation model for zero-shot speech synthesis, prioritizing the ability to generalize to unseen voices and styles from audio prompts. Besides, VoxCPM shows some implicit control ability for expressive speech synthesis. As shown in **Appendix F.5 (Fig. 3) and demopage**, even without explicit control tags, VoxCPM demonstrates emergent controllability based on input text content. The t-SNE analysis shows that the model automatically clusters prosodic styles based on text semantics (e.g., distinguishing "Poetry" from "News" reading styles), proving it captures rich expressive capabilities directly from pre-training data. Furthermore, the demopage's Context-Aware Speech Generation section provides audio samples illustrating this: *without a reference audio for continuation, VoxCPM can implicitly infer expressive style from the input text content alone*.
>
> These preliminary observations confirm the viability of extending VoxCPM toward explicit controllable synthesis in our future work. Specifically, we have conducted preliminary experiments on the Capspeech dataset [R1], validating the feasibility of generating speech styles controlled by text descriptions.
>
> [R1] Wang, Helin, et al. "Capspeech: Enabling downstream applications in style-captioned text-to-speech." arXiv preprint arXiv:2506.02863 (2025).

---

### Official Review · Reviewer_Nvjv · 2025-10-30

**Soundness:** 3
**Presentation:** 3
**Contribution:** 3
**Rating:** 8
**Confidence:** 5

**Summary:**

The paper presents VoxCPM, a hierarchical model for zero-shot TTS. The authors point out that existing discrete token approaches, while stable, tend to lose subtle acoustic nuances, whereas fully continuous models preserve detail but often become unstable over longer utterances. VoxCPM addresses this by introducing a semi-discrete differentiable bottleneck that separates stable discrete, potentially linguistic and prosodic information, from finer acoustic details, avoiding the scalability issues of large discrete codebooks while keeping the system trainable end-to-end.

The model generates speech in stages: a text-semantic language model first captures content and prosody, this representation is regularized through a scalar quantization layer to form a coarse conditioning, and a residual acoustic module restores missing details such as timbre and micro-prosody. A local diffusion transformer then synthesizes the final latent audio segments, conditioned on both text and previously generated context. The latents are derived from a causal VAE.
Training is done jointly with a flow-matching objective.

The proposed method is technically sound and well-motivated, addressing a central bottleneck in hierarchical speech synthesis. The proposed semi-discrete, differentiable quantization + residual refinement framework feels like a natural evolution of current VALL-E-style architectures toward more continuous, end-to-end differentiable systems of hierarchical speech synthesis.

**Strengths:**

1. Cohesive hierarchical design: The decomposition into semantic (TSLM+FSQ), residual (RALM), and generative (LocDiT) components is conceptually elegant and grounded in recent hierarchical TTS models.

2. Innovative FSQ usage: Using scalar quantization as a differentiable inductive bias (rather than discrete target) is novel—addresses VQ scalability while maintaining stability.

3. End-to-end differentiability: Gradients flow through quantization and all components, unlike most multi-stage systems.

4. Strong performance across benchmarks

5. The t-SNE plots of the TSLM and RALM show a clear division of roles of the two stages.

**Weaknesses:**

1. Complex multi-stage hierarchy:
The full pipeline—LocEnc → TSLM → FSQ → RALM → LocDiT—adds several interacting components and training dependencies. Despite being end-to-end, this design could be challenging to scale or fine-tune efficiently.

2. Limited analysis of latency and efficiency:
The paper emphasizes real-time feasibility via a causal VAE but provides no measured latency or throughput results. It remains unclear how the diffusion-based LocDiT performs under streaming constraints.

3. FSQ interpretability and scalability:
While the ablation results show that the FSQ layer improves stability, plays a distinct role from the RALM, and remains largely speaker- and acoustics-agnostic, the claim that FSQ representations correspond to linguistic content is not directly supported by experimental evidence. This interpretation is plausible, but could be strengthened through probing analyses—such as phoneme prediction or ASR accuracy tests—to confirm that the FSQ indeed captures linguistic or prosodic structure. In addition, the paper does not discuss how well these semi-discrete representations transfer or scale when fine-tuned on different datasets, even within the same language domain.

4. The diagram should include the mathematical notation to make it clearer which token/color corresponds to which notation in the equation and text.

**Questions:**

See weaknesses above.

---

> ### Author Response · Authors · 2025-11-24
> **Response to Reviewer Nvjv(1/2)**
>
> Thank you for your insightful comments and and strong support, particularly recognizing our work as a "natural evolution" with a "cohesive hierarchical design" and "innovative FSQ usage." We address your questions regarding interpretability, scalability and latency below.
>
> > **W1: Scalability of the Hierarchical Architecture**
>
> We sincerely appreciate your insightful comment regarding the scalability of our multi-component design (LocEnc → TSLM → FSQ → RALM → LocDiT). While the architecture introduces multiple dependencies, its end-to-end differentiability and modularity ensure practical scalability.
>
> To empirically demonstrate this, we trained larger variants of VoxCPM (1B and 3B parameters) on the Emilia dataset for 200k iterations. The results, shown below, confirm that increasing model capacity consistently improves performance, demonstrating that the architecture scales effectively. We have added this experiment in Appendix F.8 of the revised version.
>
> | Model Size | EN-WER ↓ | EN-SIM ↑ | ZH-CER ↓ | ZH-SIM ↑ |
> |------------|----------|----------|----------|----------|
> | 0.5B       | 2.98     | 62.6     | 1.77     | 70.4     |
> | 1B         | 2.95     | 65.9     | 1.82     | 72.0     |
> | 3B         | 2.60     | 66.7     | 1.78     | 72.3     |
>
>
> *Notes: VoxCPM-1B has a 28-layer TSLM (hidden-size=2048), a 7-layer RALM (hidden-size=2048) and a 6-layer  LocEnc & LocDiT (hidden-size=1024), while VoxCPM-3B has a 32-layer TSLM (hidden-size=2560), a 8-layer RALM (hidden-size=2560) and a 8-layer LocEnc & LocDiT (hidden-size=1024).*
>
> Fine-tuning is also efficient, as gradients flow holistically, and we observed no significant overhead compared to single-LM baselines. Potential challenges (e.g., memory) are mitigated by our low frame rate (12.5 Hz), enabling longer contexts.
>
> > **W2: Latency and Efficiency Analysis**
>
> We appreciate the suggestion to provide concrete efficiency and latency metrics. We have evaluated the Real-Time Factor (RTF) of VoxCPM against other TTS models on a single NVIDIA RTX 4090 GPU. VoxCPM achieves a remarkably low RTF of 0.17, significantly faster than CosyVoice2 (0.52) and SparkTTS (0.80), due to its efficient 12.5 Hz token rate and lightweight components. Throughput is ~73 tokens/s for LM backbone in VoxCPM.
>
> | TTS Model | RTF ↓ |
> |-------|-------|
> | CosyVoice 2 | 0.52 |
> | SparkTTS | 0.80 |
> | IndexTTS 2 | 0.85 |
> | VoxCPM | 0.17 |
>
> ***Streaming Synthesis Details***
>
> Furthermore, VoxCPM supports true streaming synthesis with <100ms first-packet latency theoretically: the LocDiT generates patches in <20ms under 10 iterations due to its short local context and lightweight parameters, and the causal AudioVAE can process them incrementally. For smooth playback, we buffer the last 3 latents, yielding 80ms audio chunks per step. We will release the streaming inference scripts of VoxCPM.
> We have added the RTF comparison and the streaming synthesis implementation in the Appendix F.9.
>
> [...continuing...]

---

> ### Author Response · Authors · 2025-11-24
> **Response to Reviewer Nvjv (2/2)**
>
> > **W3: FSQ Interpretability and scalability**
>
> This is an insightful question. We discuss the interpretability and scalability of the representations separately:
>
> - **Interpretability:** To strengthen evidence that FSQ captures linguistic/prosodic structure, we conducted Layer-wise Probing Experiments including PR and ASV as suggested (detailed in the **General Response**). TSLM & FSQ hidden states yield a relatively low Phoneme Error Rate (PER) and WER but higher Speaker EER compared to acoustic embeddings and RALM hidden states. This empirically confirms our hypothesis: the FSQ bottleneck indeed restricts TSLM to focus more on the **linguistic-content information** and filters out some acoustic details. We have add this analysis in the Appendix F.7 of the revised version.
>
> - **Scalability/Transfer**: We appreciate the reviewer's insightful question regarding the transferability and scalability of the semi-discrete representations.
>     - **Capacity Analysis**: We initially investigated the necessary capacity of the bottleneck through FSQ dimension ablation studies (detailed in Table 4), which helped us determine appropriate setting (128-d or 256-d) for the Emilia dataset. This confirms that the FSQ dimension is a configurable hyperparameter that regulates the capacity of the semi-discrete space required to model the necessary semantic and prosodic information.
>     - **Empirical Scaling**: Building on the above ablation study, we adopted the 256-d FSQ configuration for our internal 1M-hour dataset. The strong performance achieved by VoxCPM on the large-scale dataset (Table 1) empirically validate that this configuration transfers effectively, suggesting that the 256-d FSQ provides a suitable balance between capacity and constraint for diverse data scales. We would like to provide an intuitive interpretation: *this balance allows sufficient capacity for complex semantic-prosodic variations while maintaining the stabilizing regularization effect*. We sincerely thank the reviewer for highlighting the importance of systematic exploration in this area, and we recognize that more comprehensive investigation of the relationship between data characteristics and optimal bottleneck design would be valuable for future work.
>
> > **W4: Mathematical Notation in Diagrams**
>
> Thank you for this helpful feedback. We have revised Figure 1 to include clear mathematical notations matching the equations and text.

---

> > ### Comment · Reviewer_Nvjv · 2025-11-27
> > **Response to Author Comments**
> >
> > Thank you for the detailed clarifications and additional experiments. I am satisfied. I will keep my positive rating of the paper.

---

### Official Review · Reviewer_kJR8 · 2025-10-31

**Soundness:** 2
**Presentation:** 3
**Contribution:** 3
**Rating:** 4
**Confidence:** 3

**Summary:**

The paper introduces VoxCPM, a framework that addresses speech expressivity and intelligibility challenges. Codec-based methods enable stable autoregressive generation but lose fine-grained acoustic details due to quantization, while continuous approaches accumulate errors over long sequences. VoxCPM combines a Text-to-Speech Language Model (TSLM) for generating semantic and prosodic plans with a Residual Acoustic Language Model (RALM) that restores fine acoustic details. Guided by these components, a local diffusion-based decoder produces high-fidelity speech latents. Experiments on two datasets show that VoxCPM achieves better results with fewer model parameters.

**Strengths:**

- The paper proposes a differentiable FSQ bottleneck that divides semantic–prosodic planning from acoustic rendering, maintaining full end-to-end trainability and overcoming the limits of both discrete and continuous methods.

- The paper reduces computational cost by using a VAE-based model that works in a compact latent space rather than on raw waveforms. A causal design allows for low-latency, real-time streaming synthesis.

- The paper shows that VoxCPM outperforms other open-source models on the experimented dataset using 0.5B parameters, outperforming larger models such as IndexTTS2 (1.5B) and HiggsAudio-v2 (3B).

**Weaknesses:**

- The paper combines existing components (pre-trained LLM, FSQ, diffusion decoder) without explaining why this combination works beyond T-SNE visualizations. What specific features do TSLM vs. RALM capture? Missing are probing experiments, attention analysis, layer-wise studies, or comparisons with alternative quantization methods. The paper should provide a few fundamental insights to verify their claims.

- The model does not use any explicit alignment mechanism, such as attention or duration modeling, to synchronize text tokens with acoustic frames. Instead, it depends on implicit correlations learned during training, which can result in mispronunciations, timing instability, or prosodic drift in longer utterances.

- The ablation studies don’t justify critical design decisions, including the 24-layer/6-layer split between TSLM and RALM, the contribution of pre-trained LLM initialization quality (e.g., comparing MiniCPM-4 vs. other pre-trained models), and the necessity of the FSQ bottleneck itself by comparing against simpler hierarchical architectures without quantization.

**Questions:**

- Regarding the acoustic embeddings (E<i), what specific acoustic features (pitch variations, timbre, voice quality) does it learn?

- Could the author clarify how the proposed framework manages long audio sequences?

- RALM is reported to recover “speaker identity” implicitly, yet the model has no explicit conditioning on speaker embeddings or style tokens. Could the authors clarify how speaker identity is actually captured?

---

> ### Author Response · Authors · 2025-11-24
> **Response to Reviewer kJR8 (1/2)**
>
> Thank you for your constructive feedback and for recognizing that VoxCPM "overcomes the limits of both discrete and continuous methods" with efficiency and strong results. We address your concerns below with new experimental evidence and detailed clarifications, which have been incorporated into the revised manuscript.
>
> > **W1 & Q1: Interpretability and Feature Analysis**
>
> The combination works by structurally separating tasks via FSQ's inductive bias, allowing TSLM to focus on semantic-prosodic planning and RALM on acoustic rendering. We agree that quantitative evidence is necessary to move beyond t-SNE visualizations and fully explain the task specialization of TSLM and RALM.
>
> As detailed in the **General Response**, we conducted Layer-wise Probing Experiments including PR, ASR, and ASV based on the SUPERB benchmark. The results quantitatively confirm our architectural hypothesis:
>  - **TSLM Hidden States**: Achieved the lowest PER and WER and relatively high EER, validating its specialization in capturing the **linguistic content information (semantic-prosodic plan)**.
> - **RALM Hidden States**: Achieved the lowest Speaker EER and relatively high WER/PER, validating its role in restoring **speaker identity and fine-grained acoustic residuals**.
> - **Acoustic Embeddings ($E_{<i}$)**: The acoustic embeddings are derived from the LocEnc, which compresses the low-level, continuous acoustic latents from AudioVAE. Our probing results show that the acoustic embeddings contain **weak semantic information and carries fundamental acoustic features like timbre**. Furthermore, the TSLM subsequently reduces the speaker information from this input, indicating that the TSLM utilizes $E_{<i}$ for time-domain context but prioritizes semantic modeling over acoustic identity preservation.
>
> > **W2 & Q2: Implicit Alignment and Long-Form Speech**
>
> - **Implicit Alignment and Stability:** We acknowledge that LLM-based TTS relies on implicit alignment. While this paradigm is the dominant trend due to its quality and scalability, we agree with the reviewer that challenges like mispronunciation, timing instability, or prosodic drift can potentially emerge in very long sequences. However, compared to purely continuous models, our FSQ bottleneck serves as an effective regularizer, which significantly constrains the TSLM's output space and thus helps to alleviate the accumulation of **"prosodic drifting"** and **"time instability"** errors. As a result, VoxCPM achieves a state-of-the-art WER among open-sourced AR systems, demonstrating that our method learns a relatively robust alignment that is resilient to **mispronunciations**.
> - **Long-Form Speech Synthesis**: Our architecture is specifically optimized for long-form synthesis through an **ultra-low token rate**. VoxCPM operates at 12.5 Hz in the LM backbone, which is significantly more compact than standard codecs (typically 25-75 Hz). This low rate allows the VoxCPM to generate  minute-level speech efficiently like VibeVoice [R1] and FireRedTTS2 [R2]. For indefinitely long generation, the inherent causal and AR design supports a standard chunk-based streaming strategy to maintain stability.
>
>
> [...continuing...]

---

> > ### Author Response · Authors · 2025-11-24
> > **Response to Reviewer kJR8 (2/2)**
> >
> > > **W3: Ablation Studies for Architectural Decisions**
> >
> > We provide clarification and references for the critical design decisions questioned:
> > - **Layer Split (24/6):** This asymmetry is an empirical allocation **grounded in prior hierarchical models** like SpearTTS [R3], Qwen-Omni [R4], and LeVo [R5], reflecting the distinct task complexity. Semantic modeling (TSLM) requires deep, pre-trained hierarchies (24 layers) to capture complex linguistic and seman knowledge, while acoustic refinement (RALM) is a conditional residual mapping task, efficiently solved with a shallower network (6 layers). We followed this empirical setup to impelment VoxCPM.
> > - **FSQ Necessity:** The necessity of the quantization bottleneck is addressed by the original ablation: **"w/o FSQ: d1024s$\infty$" (Table 4, Row 5)**. This setting removes the quantization step while retaining the hierarchical structure. Compared to the default setting, removing FSQ leads to significant degradation in WER/CER, demonstrating that without the quantization bottleneck, the continuous hidden states suffer from task entanglement, validating FSQ as a necessary component.
> > - **Pre-training Initialization:** To prove the role of LLM parameter initialization, we added an ablation study (also included in Table 4 and Section 4.4 of the revised verson). It can be found that removing pre-trained text LM initialization results in a significant degradation in intelligibility (e.g., EN-WER 2.98 vs. 5.24). This confirms that the pre-trained knowledge is critical for establishing the TSLM's initial capability and stability. Of course, we believe that the gains derived from initializing the TSLM with weights from other pre-trained LLMs are similar, we just chose MiniCPM-4 as the TSLM backbone, which is not the focus of the method. We observe that models trained from random initialization shows a marginal increase in SIM, suggesting that without the strong linguistic guidance from the pre-trained LLM, the model might implicitly allocate more capacity to acoustic feature modeling.
> >
> >
> > | Model Setting | EN-WER ↓ | EN-SIM ↑ | ZH-CER ↓ | ZH-SIM ↑ | Hard-CER ↓ | Hard-SIM ↑ |
> > |-----------|--------|--------|--------|--------|--------|--------|
> > | default setting: LM init. in TSLM | 2.98 | 62.6 | 1.77 | 70.4 | 18.19 | 64.9 |
> > | w/o LM init. in TSLM| 5.24 | 63.4 | 2.41 | 70.9 | 24.66 | 65.6 |
> >
> >
> > > **Q3: Speaker Modeling in RALM**
> >
> > We clarify that VoxCPM is a foundation model that achieves zero-shot timbre and style modeling implicitly, without relying on explicit speaker embeddings or style tokens.
> >
> > The RALM's ability to recover or infer speaker identity stems from two primary mechanisms driven by its architecture:
> > - **In-Context Learning (Zero-Shot Speech Continuation)**
> >     - This is the core mechanism for zero-shot speaker/style modeling. When provided with an arbitrary audio prompt, VoxCPM operates in the speech continuation mode.
> >     - Specifically, the RALM utilizes the prompt's acoustic latent history ($\mathbf{E}_{<i}$) to perform ICL, successfully modeling and extending the provided timbre and style of the reference speech.
> >     - Crucially, due to its Autoregressive (AR) nature, the RALM uses the generated features from the previous acoustic patch to condition the current step, inherently ensuring long-term consistency of the speaker's timbre and style across the output sequence.
> > - **Inference from Text (Context-Aware Speech Generation)**
> >     - Without an audio prompt, the RALM must sample the first several speech latents from the internal distribution to initiate synthesis. As shown in Figure 1, the RALM receives the TSLM's text hidden states as sequence conditioning (Eq. 4).
> >     - This structural design grants the RALM the potential to model the association between the text content and the relevant speech style. Through training on vast amounts of diverse real-world data, the model learns these implicit associations.
> >     - Therefore, even without a reference audio prompt, the model is capable of spontaneously inferring and sampling a suitable style and timbre based on the input text content (e.g., predicting a "news style" for news content or a "casual tone" for conversational text). Our tests confirm that this sampling process is not entirely random.
> >
> > [R1] Peng, Zhiliang, et al. "Vibevoice technical report." arXiv preprint arXiv:2508.19205 (2025).
> >
> > [R2] Xie, Kun, et al. "Fireredtts-2: Towards long conversational speech generation for podcast and chatbot." arXiv preprint arXiv:2509.02020 (2025).
> >
> > [R3] Kharitonov, Eugene, et al. "Speak, read and prompt: High-fidelity text-to-speech with minimal supervision." Transactions of the Association for Computational Linguistics 11 (2023): 1703-1718.
> >
> > [R4] Xu, Jin, et al. "Qwen2. 5-omni technical report." arXiv preprint arXiv:2503.20215 (2025).
> >
> > [R5] Lei, Shun, et al. "LeVo: High-Quality Song Generation with Multi-Preference Alignment." arXiv preprint arXiv:2506.07520 (2025).

---

### Official Review · Reviewer_emjo · 2025-11-01

**Soundness:** 3
**Presentation:** 4
**Contribution:** 2
**Rating:** 4
**Confidence:** 5

**Summary:**

The paper introduces a hierarchical AR framework based on semi-discrete residual representations to mitigate the trade-off between stability and expressivity observed in recent LLM-based TTS systems using speech tokenizers. The proposed architecture comprises a Text-Semantic Language Model that captures high-level linguistic and prosodic structure, and a Residual Acoustic Language Model that reconstructs fine-grained acoustic details. It further incorporates a differentiable FSQ bottleneck with LocDiT and a causal VAE to achieve unified and high-fidelity speech generation.

**Strengths:**

- The paper presents a clear and well-structured framework that effectively balances stability and expressivity through hierarchical semantic-acoustic modeling. The proposed combination of TSLM, RALM, and the FSQ bottleneck is conceptually coherent and technically sound.
- The writing is clear and easy to follow, and the experimental results are well organized and convincingly support the main claims.

**Weaknesses:**

- The paper introduces FSQ as a semi-discrete bottleneck, but its function appears closer to latent feature regularization than true discrete representation learning. Because FSQ is not used for token prediction, it mainly constrains feature space rather than modeling discrete structure. A comparison with VAE-based latent regularization would help determine whether FSQ provides meaningful representational advantages or simply smoother feature constraints.
- The authors claim the system as an  end-to-end framework, but it cannot be considered truly  end-to-end because it relies on a pre-trained audio VAE to extract continuous speech latents.
- The paper provides limited details about the Causal Audio VAE, even though the design of latent speech tokens can significantly affect overall performance.

**Questions:**

Q1. Does the observed separation in Appendix F.5 - Figure 2 truly result from the FSQ bottleneck, or is it primarily due to the pre-trained initialization of the semantic LLM? How would the t-SNE visualization change if the TSLM were trained from random initialization?

Q2. In most recent LLM-based TTS systems, increasing model capacity generally improves both speaker similarity and intelligibility. However, in the ablation study (Table 4), deeper TSLM layers slightly increase SIM but noticeably degrade WER. It would be helpful if the authors could clarify the reason for this. observation. Specifically, how can the effects of structural disentanglement be distinguished from capacity related trade-offs, and might the observed degradation reflect limitations in training dynamics rather than an inherent need for the RALM module?

Q3. The paper states that the TSLM uses MiniCPM-4-0.5B as its backbone, while the overall model size is also reported as 0.5B parameters. Could the authors clarify how this total is computed? Specifically, how many parameters are allocated to the AudioVAE and the RALM modules?

---

> ### Author Response · Authors · 2025-11-24
> **Response to Reviewer emjo (1/2)**
>
> We sincerely appreciate your thoughtful and constructive feedback, and we are grateful that you recognize VoxCPM as "conceptually coherent and technically sound."  We have conducted new experiments and provided detailed clarifications to address your concerns below.
>
> > **W1: FSQ vs. VAE-based Regularization**
>
> You raised an insightful point: does the FSQ bottleneck provide a meaningful  representational advantage over continuous latent regularization, such as that provided by a VAE? We acknowledge that FSQ primarily constrains the feature space. However, it provides a **rigid, quantization-based constraint** that enforces task separation more effectively than continuous methods, without suffering from VAE optimization trade-offs (e.g., posterior collapse, when KL loss weight is large).
>
> To test this, we replaced the FSQ bottleneck with a **continuous VAE bottleneck**. We specifically chose a dimension of 16 for the continuous bottleneck (16d-VAE), as high-dimensional continuous latents (e.g., 256d) empirically suffer from posterior collapse or inefficient compression due to optimization challenges. For a fair comparison, we also provided a 16d-FSQ model result to show the effect of quantization at the same dimensionality.
>
> The results (included in Appendix F.3) clearly demonstrate the superior stability provided by the FSQ:
>
> - **Impact of Quantization (16d-VAE vs. 16d-FSQ):**  Under the same 16-dimensional setting, the FSQ model achieves better robustness than the VAE. The VAE bottleneck nearly doubles the error rate on hard cases compared to the FSQ  (28.17 vs. 14.42). This confirms that the **semi-discrete nature** of FSQ provides an indispensable inductive bias for stability that continuous regularization lacks.
> - **Capacity vs. Stability:** While the continuous 16d-VAE achieves slightly higher English speaker similarity than 16d-FSQ (likely because continuous vectors can encode finer acoustic details), this comes at the cost of **intelligibility**.
> - **Overall Performance:** Our default 256d-FSQ strikes the best balance, maintaining high expressivity without the optimization instability associated with high-dimensional continuous bottlenecks.
>
> | Model Setting | EN-WER ↓ | EN-SIM ↑ | ZH-CER ↓ | ZH-SIM ↑ | Hard-CER ↓ | Hard-SIM ↑ |
> |------------|----------|----------|----------|----------|------------|------------|
> | default: 256d-FSQ | 2.98 | 62.6 | 1.77 | 70.4 | 18.419 | 64.9 |
> | 16d-VAE | 3.56 | 62.1 | 1.94 | 69.7 | 28.17 | 62.7 |
> | 16d-FSQ | 3.22 | 60.4 | 1.87 | 70.5 | 14.42 | 66.2 |
>
> > **W2 & W3: End-to-End Definition and AudioVAE Details**
>
> We appreciate the opportunity to provide more precise terminology.
>
> - **End-to-End Definition**: While VoxCPM is a hierarchical architecture, its design ensures it is more "end-to-end" than traditional multi-stage cascaded systems (e.g., CosyVoice) which are typically trained disjointly. VoxCPM is **holistically optimized** because gradients propagate from the final diffusion loss through the entire chain (LocEnc→TSLM→RALM→LocDiT). And we achieve stable and expressive speech generation by modeling directly on **continuous speech latents** in the LM backbone. We clarify that our paper's core claim is that we eliminate the dependency on external discrete speech tokenizers (e.g., S3Tokenizer), which impose a severe, fixed information ceiling. We acknowledge the AudioVAE as a continuous speech tokenizer. Our system is considered end-to-end differentiable from text input to the speech latent space.
> - **AudioVAE Details**: We would like to clarify that the details of the AudioVAE has already been introduced in **Appendix D.1**. We have added a clear statement in Section 3.5 to indicate it.
>
> > **Q1: Separation driven by Architecture or Initialization**
>
> This is an insightful question that probes the origin of our structural disentanglement. To rigorously decouple the effects of the hierarchical architecture from the pre-trained LM weights, we conducted a controlled experiment: we trained a VoxCPM variant with **fully random TSLM initialization** and performed a new t-SNE visualization of its latent spaces.
>
> The **new visualization results (included in Appendix F.5)** reveal a critical finding: *Even without the benefit of pre-trained knowledge, the latent space exhibits the similar fundamental division of labor.* The TSLM-FSQ hidden states still show clear semantic relevance and weak speaker relevance, while the RALM hidden states display strong speaker-dependent clustering.
> Crucially, while this self-learned specialization is slightly weaker compared to the model initialized with pre-trained LM, the presence of this architecture still shows that the residual semi-discrete bottleneck design can lead to a certain degree of semantic-acoustic division of labor. The pre-trained LM weights then serve to significantly sharpen, stabilize, and enhance this inherent modeling capacity. We have also incorporated this analysis into Appendix F.5.
>
> [...continuing...]

---

> ### Author Response · Authors · 2025-11-24
> **Response to Reviewer emjo (2/2)**
>
> > **Q2: Clarification on Ablation Study and Model Capacity**
>
> We apologize for the confusion regarding Table 4. The performance degradation in the reported "w/o RALM: TSLM (30 layers)" setting was primarily due to initialization, not model capacity. The corrected definition of these two settings are:
>
> - **w/o RALM: TSLM(24 layers)→LocDiT**: The TSLM was fully Initialized from the pre-trained MiniCPM-4-0.5B (24 Layers). We renamed it as **"w/o RALM:TSLM(24 layers, LM init.)→LocDiT"**.
> - **w/o RALM: TSLM(30 layers)→LocDiT**: The TSLM was fully random initialization since no 30-layer MiniCPM-4 existed, thus leading to relatively poor intelligibilty. We renamed it as **"w/o RALM:TSLM(30 layers, random init.)→LocDiT"**.
>
> Therefore, the degradation observed in the original 30-layer TSLM experiment was simply due to the loss of pre-trained knowledge (random initialization), not an inherent limitation of model depth. We performed extensive new ablations to isolate the effects:
> - **w/o RALM:TSLM(30 layers, partial init.)→LocDiT**: A 30-layer TSLM where the first 24 layers are initialized from MiniCPM and the last 6 are random, without hierarchical  design.
> - **Hierarchical, w/o LM init. in TSLM**: The proposed hierarchical framework consisent with VoxCPM, but with the TSLM randomly initialized.
>
> | Model Setting | EN-WER ↓ | EN-SIM ↑ | ZH-CER ↓ | ZH-SIM ↑ | Hard-CER ↓ | Hard-SIM ↑ |
> |--------------|--------|--------|--------|--------|--------|--------|
> | default setting (Hierarchical, LM init.) | 2.98 | 62.6 | 1.77 | 70.4 | 18.19 | 64.9 |
> |w/o RALM:TSLM(24 layers, LM init.)→LocDiT| 4.34 | 61.8 | 3.05 | 69.4 |25.00| 63.8|
> | w/o RALM:TSLM(30 layers, random init.)→LocDiT | 5.35 | 62.6 | 3.46 | 69.8 | 30.40 | 63.9 |
> | w/o RALM:TSLM(30 layers, partial LM init.)→LocDiT | 4.12 | 62.0 | 3.07 | 69.6 | 26.20 | 63.1 |
> | Hierarchical, w/o LM init. in TSLM | 5.24 | 63.4 | 2.41 | 70.9 | 24.66 | 65.6 |
>
> As shown above, we can get these conclusions:
> - **Capacity vs. Hierarchical Structure**: By comparing single-stream models with the same initialization foundation, that is, [w/o RALM:TSLM(24 layers, LM init.)→LocDiT] vs. [w/o RALM:TSLM(30 layers, partial LM init.)→LocDiT], we demonstrated that increasing capacity yielded only marginal gains (e.g., EN-WER from 4.34 to 4.12). Conversely, the Hierarchical VoxCPM (default setting) significantly outperforms all the single-stream models (EN-WER 2.98). This confirms that **structural separation (TSLM/RALM) via FSQ bottleneck** is the dominant factor for performance, not merely increased capacity.
> - **Inherent Architectural Advantage:** When comparing the two models trained from completely random initialization, that is, [Hierarchical, w/o LM init. in TSLM] vs. [w/o RALM:TSLM(30 layers, random init.)→LocDiT], we found that the hierarchical architecture of VoxCPM achieves superior performance across both intelligibility and speaker similarity. Specifically, It yields lower error rates (EN-WER: 5.24 vs. 5.35 and ZH-CER: 2.41 vs. 3.46) , alongside higher speaker similarity (EN-SIM: 63.4 vs. 62.6; ZH-SIM: 70.9 vs. 69.8) compared to the single-stream counterpart. This simultaneous improvement confirms the **intrinsic benefit of the hierarchical residual design**: it encourages a specialized division of labor—allowing the RALM to focus effectively on fine-grained acoustic reconstruction while the TSLM handles semantic structure—resulting in superior generation quality, even **without the strong semantic grounding from pre-trained parameters**.
>
> We have added the detailed analysis into Section 4.4 of the revised version.
>
>
>
> > **Q3: Parameter Amount and Difinition**
>
> We appreciate the request for clarification on the parameter count. We follow the common naming convention in recent LLM-based TTS research (e.g., CosyVoice2-0.5B which has 0.5B LM and 0.1B FM, HiggsAudio-3B which has 3.6B LM and 2.2B Audio Dual FFN), where the labeled size refers to the core generative autoregressive parameters, excluding auxiliary components like encoders / decoders / embeddings.
>
> The breakdown for VoxCPM-0.5B is:
>
> - **TSLM**:  433M parameters (MiniCPM-derived).
> - **RALM**:  89M parameters.
> - LocDiT: 64M parameters.
> - LocEnc: 59M parameters
> - StopPredictor: 1M parameters
> - FSQ:  0.5M parameters
>
> The total core AR parameter count for VoxCPM-0.5B is approximately **522M (as TSLM + RALM)**, comprising the primary generative components responsible for global sequence processing.
> Besides, the AudioVAE (75M) is treated as a external frozen tokenizer, which is also not included in this count.
> We have added the detailed parameter information in Table 5 of the revised version.

---

### Official Review · Reviewer_ada6 · 2025-11-02

**Soundness:** 3
**Presentation:** 3
**Contribution:** 3
**Rating:** 6
**Confidence:** 5

**Summary:**

This manuscript proposes a novel end-to-end TTS framework, VoxCPM, to address the fundamental trade-off between discrete and continuous speech representations, where discrete representations maintain stability but sacrifice expressivity, while continuous representations retain acoustic richness but are prone to error accumulation. The hierarchical semantic-acoustic modeling framework is well-motivated and addresses some limitations in current TTS systems.

The proposed VoxCPM is with the hierarchical semantic-acoustic architecture, consisting of a Text-Semantic Language Model (TSLM), a Residual Acoustic Language Model (RALM), and the Local Diffusion Transformer (LocDiT). The key innovation is the differentiable Finite Scalar Quantization bottleneck, which serves not as a prediction target, but as an intermediate regularization mechanism to induce natural task separation.

The manuscript demonstrates that their 0.5B-parameter model achieves state-of-the-art zero-shot TTS performance with extensive experiments and ablations.

**Strengths:**

1. The major contribution, using a differentiable FSQ bottleneck as an internal inductive bias rather than a predictive target, is a elegant and well-motivated solution to the task entanglement problem. By regularizing the hidden state, it forces the TSLM to focus on stable semantic-prosodic structures while offloading fine-detail modeling to the RALM.

2. The paper provides compelling quantitative results. VoxCPM achieves state-of-the-art performance among open-source models on two challenging benchmarks, SEED-TTS-EVAL and CV3-EVAL. The ablation studies conclusively validate the authors' core hypotheses.

3. While the best-performing model (VoxCPM) is trained on a large, proprietary dataset , the authors have responsibly run all critical ablation studies on the public Emilia dataset.

**Weaknesses:**

1. While the empirical results are strong, the paper lacks theoretical justification for why FSQ specifically induces semantic-acoustic disentanglement

2. No formal analysis of the "quantization ceiling" or error accumulation mechanisms, and how the proposed architechture can effectively address them.

3. The claim that FSQ acts as an "inductive bias" needs more rigorous theoretical grounding and mathematical proof.

4.  The main SOTA results are from the VoxCPM model trained on a 1-million-hour internal, bilingual dataset. While the authors provide VoxCPM-Emilia, trained on a public dataset, the top-line performance is dependent on this massive, inaccessible dataset. This limits the direct reproducibility of the primary claims, although the architectural insights from the Emilia-based ablations remain sound. Some baselines (F5-TTS, GPT-Sovits) may use significantly less data, making direct comparison problematic.

5. The paper doesn't discuss whether the performance gains are primarily from architecture or data scale

6. The entire model generates latents from a pre-trained Causal VAE. This VAE is itself a form of compression. The paper criticizes discrete tokenizers for their "quantization ceiling", but doesn't discuss the potential information bottleneck or quality ceiling imposed by its own VAE. The quality of this VAE is critical to the final output, but it's only briefly detailed.
The author also claimed that the voxCPM do not rely on a external tokenizer. However the VAE is itself a tokenizer.

**Questions:**

1. You sum TSLM and RALM outputs. Did you experiment with other fusion strategies (concatenation, gating, attention)?

2. The paper states FSQ is "analogous to the first layer of Residual Vector Quantization (RVQ)". Did you experiment with using a learned vector quantizer (e.g., a single VQ layer) as the bottleneck instead of the non-learned FSQ? It seems a learned codebook might create an even more efficient semantic-prosodic skeleton. And what happens if you replace FSQ with VQ-VAE or other quantization methods?

3. Please discuss the potential quality ceiling imposed by their pre-trained causal VAE  and how it relates to the "quantization ceiling" they criticize in discrete tokenizers.

4. Please try your best to provide more theoretical analysis as mentioned in "Weaknesses".

4. The VAE is also external tokenizer.

5. For citation formatting, please ensure consistent citation style (some entries lack year information)

---

> ### Author Response · Authors · 2025-11-24
> **Response to Reviewer ada6 (1/2)**
>
> Thank you for your comprehensive review and for recognizing VoxCPM as an "elegant and well-motivated solution" to the task entanglement problem. We address your concerns below with further explanation and experimental evidence.
>
> > **W1,W3 & Q4: Interpretability for FSQ as Inductive Bias**
>
> We sincerely appreciate your suggestion for a **deeper theoretical and empirical justification** of the FSQ-induced semantic-acoustic modeling. To address this fundamental insight into our architecture, we provide a unified explanation based on the Information Bottleneck (IB) Principle and validation from new probing experiments:
> - **Theoretical Grounding (Information Bottleneck)**: The FSQ layer acts as a capacity-constrained bottleneck between the high-capacity TSLM and the downstream RALM. Mathematically, the scalar quantization step inherently acts as a lossy compression filter. To minimize the final end-to-end diffusion loss, the TSLM is heavily penalized if it attempts to encode high-variance, fine-grained acoustic details (e.g., specific residuals or timbre), as this information would be severely distorted by the fixed, scalar quantization. Consequently, the TSLM is naturally forced to prioritize low-variance, stable features—namely, the semantic and global prosodic structure. This mechanism effectively offloads the modeling of high-frequency, complex acoustic residuals to the RALM (added in **Appendix G.1**).
> - **Empirical Validation (Probing)**: Our layer-wise probing results (see **General Response**) empirically validate this induced division of labor. TSLM representations demonstrate high semantic accuracy (PER=45.60%) but are acoustic-agnostic (EER=18.70%), while RALM exhibits the opposite behavior (PER=53.94%, EER=13.24%), confirming the FSQ's effectiveness as an internal inductive bias for hierarchical task separation.
>
> > **W2, W6, Q3 & Q5: VAE and the Quantization Ceiling**
>
> Thank you for pointing out the conceptual role of the AudioVAE as a continuous speech tokenizer. We acknowledge this clarification and have updated the manuscript to provide a more precise description, emphasizing that our system eliminates dependency on external discrete speech tokenizer (like VQ-based Codecs).
>
> The "quantization ceiling" we criticize is as follows:
>
> - **Continuous Latents vs. Discrete Codebooks**: Discrete tokenizers like S3Tokenizer map speech signals to a fixed, finite codebook of size K,imposing a rigid, finite ceiling on expressive information (the "quantization ceiling"). Our AudioVAE operates in a 64-dimensional continuous latent space, which retains significantly richer acoustic detail and avoids the critical precision loss inherent in mapping to a finite, low-dimensional set.
> - **Internal FSQ Bottleneck Capacity**: Inside VoxCPM, the FSQ bottleneck serves as a regularizer, not a prediction target for a small codebook. We use a high-dimensional FSQ (e.g., d256s9), offering a vast representational space far exceeding traditional VQ/FSQ codebooks. Our FSQ ablation study (Table 4) shows that when we constrain the FSQ capacity to mimic a traditional small codebook (e.g., FSQ d4s9, equivalent to 6561 vocab), the clone similarity (SIM) significantly degrades, proving that full information preservation is essential for the model's performance ceiling.
> - **VAE Reconstruction Quality Ceiling**: We agree that the VAE quality is crucial. Our VAE reconstruction evaluation on Seed-TTS-Eval shows minimal degradation from the original ground truth, demonstrating that the AudioVAE imposes only a minor, acceptable bottleneck, sufficient for high-fidelity synthesis. We have added this result in Appendix G.2.
>
> |      | mel loss ↓ | stoi ↑ | WER ↓ | SIM ↑ | DNSMOS ↑ |
> |------|----------|------|-----|------|--------|
> | GT   | 0.       | 1.00  | 1.21 | 75.5 | 3.96   |
> | Recon | 0.924 | 0.948 | 1.33 | 73.6 | 3.92   |
>
> [...continuing...]

---

> ### Author Response · Authors · 2025-11-24
> **Response to Reviewer ada6 (2/2)**
>
> > **W4 & W5: Experimental Fairness: Architecture vs. Data**
>
> Thank you for your valuable suggestion, we agree that it is important to distinguish between architectural novelty and data scale dependency. Thus we have **updated Table 1 to explicitly show the training data scale of the baselines** for better comparison.
>
> When compared VoxCPM-Emilia (trained on public data, 100K hours) against competitive AR models trained on similar data scales (e.g., CosyVoice2, SparkTTS, FireRedTTS), our model consistently outperforms them in stability (WER) and similarity (SIM). While some NAR models like MaskGCT achieve higher objective metrics, AR models typically excel in prosodic naturalness and expressiveness. As shown in Table 3, evaluators rated VoxCPM and IndexTTS 2 more favorably on naturalness and similarity than MaskGCT.
>
> The strong performance of VoxCPM-Emilia confirms that our improvements stem from the **hierarchical residual autoregressive architecture**. While scaling to our internal dataset further enhances robustness and expressiveness, the architectural innovation provides the foundational performance gain.
>
> We have added this analysis in Section 4.2 in the revised version.
>
> > **Q1: Fusion Strategies.**
>
> Thank you for the suggestion. We explored various fusion strategies for combining TSLM and RALM outputs, including concatenation, gating, and attention mechanisms.
>
> As below, while the concatenation strategy achieves a marginally higher similarity score (SIM), the simply **sum** strategy performs best in terms of content intelligibility (WER/CER) .
>
> The choice of summation is deliberate: Concatenation, by doubling the feature dimension, offers the LocDiT decoder more capacity to reconstruct two independent feature streams, potentially leading to the slight SIM advantage. However, Summation is conceptually consistent with the "residual" design principle: it forces the RALM to explicitly model the residual acoustic details missing from the TSLM output. This mechanism provides the optimal balance of stability, robustness , and parameter efficiency, making it the superior architectural choice.
>
> | Fusion Strategy | EN-WER ↓ | EN-SIM ↑ | ZH-CER ↓ | ZH-SIM ↑ |
> |-----------------|----------|----------|----------|----------|
> | Sum (default)   | 2.98     | 62.6     | 1.77     | 70.4     |
> | Concatenation   | 3.16     | 64.1     | 2.88     | 71.3     |
> | Gating          | 3.75     | 63.6     | 2.04     | 70.2     |
> | Attention       | 4.92     | 59.9     | 5.43     | 66.5     |
>
> We have added this ablation study in Appendix F.6.
>
> > **Q2: FSQ & VQ Comparison**
>
> In our preliminary experiments, we attempted to use a standard learnable Vector Quantization (VQ) layer as the bottleneck. However, optimizing VQ was challenging in our high-capacity, end-to-end setting:
>
> - **Optimization Instability:** Matching the high capacity of our FSQ bottleneck (which avoids a ceiling) requires a massive, learned codebook. Optimization of such large codebooks, especially with the unstable gradient flow from the downstream diffusion loss, led to severe codebook collapse and catastrophic model failure.
> - **Stability and Differentiability:** Our FSQ bottleneck, while non-learnable in codebook values, provides the necessary high-capacity, semi-discrete constraint without the optimization pitfalls of a large, learned VQ codebook. This stability is paramount for robust end-to-end training of our hierarchical residual framework.
>
> > **Q6: Citation Formatting**
>
> Thank you for the reminder. We have revised all references to ensure a consistent citation style, adding missing years information.

---

### Author Response · Authors · 2025-11-24
**General Response (1/2)**

We sincerely thank all the reviewers for their time and valuable feedback on our submission. We are greatly encouraged that the reviewers have reached a consensus on the core value of our work, specifically acknowledging the following strengths:
- **Innovative Architecture and Conceptual Elegance** : The hierarchical framework was praised as a "conceptually elegant" and "technically sound" design (Nvjv, emjo) that effectively balances the fundamental stability-expressivity trade-off.
- **Novelty of FSQ as Inductive Bias**: The use of the differentiable FSQ bottleneck as an internal regularization mechanism was recognized as an "elegant and well-motivated solution" to task entanglement (ada6, MsWh), leading to the desired separation of semantic-prosodic planning from acoustic rendering (kJR8).
- **Strong Empirical Performance and Efficiency** : Our 0.5B-parameter model achieves state-of-the-art zero-shot TTS performance among open-source systems (kJR8, Nvjv) with "compelling quantitative results" (ada6), validating its practical strength and efficiency.
- **Clear Presentation** : Reviewers commended the manuscript for being "clear and easy to follow," well-motivated, and having excellent presentation (emjo, ada6, Nvjv).

We have polished the paper and added extensive new experiments and analysis in the revised version (major updates highlighted in **blue**):
**1. New Experiments & Ablations**
- We have added a ablation study on pre-trained LM Initialization, including new variants (e.g., partial LM init, w/o LM init in TSLM) in Section 4.4 and Table 4 (emjo, kJR8).
- We have added an ablation study comparing FSQ vs. VAE-based regularization in the bttleneck of VoxCPM in Appendix F.3 (emjo).
- We have added new t-SNE visualization of latent spaces with random TSLM initialization in Appendix F.5 (emjo).
- We have added an ablation study on TSLM/RALM hidden fusion strategies in Appendix F.6 (ada6).
- We have added layer-wise probing experiments (PR, ASR, ASV) for VoxCPM internal hidden states in Appendix F.7 (ada6, kJR8, Nvjv).
- We have conducted a scalability experiment (1B and 3B variants) in Appendix F.8 (Nvjv).
- We have provided AudioVAE reconstruction quality metrics in Appendix G.2 (ada6).

**2. Clarifications and Analysis**
- We have integrated a detailed analysis of negative metrics in Section 4.2 (MsWh).
- We have added RTF comparison and streaming synthesis details in Appendix F.9 (Nvjv).
- We have provided a theoretical grounding (information bottleneck principle) for VoxCPM in Appendix G.1 (ada6).
- We have clarified the difference between our internal FSQ bottleneck and the external VQ-based "quantization ceiling" in Appendix G.2 (ada6).
- We have updated to claim eliminating dependency on external discrete speech tokenizers in the manuscript (ada6, emjo).
- We have revised Figure 1 to include clear mathematical notations matching the equations and text (Nvjv).
- We have updated Table 1 to explicitly show the training data amount and added analysis distinguishing architecture vs. data scale in Section 4.2 (ada6).
- We have added note on evaluated model selection criterion for subjective evaluation in Table 3 caption (MsWh).
- We have clarified ablation definitions in Table 4, correcting the model setting  name (emjo).
- We have updated the parameter details of each module in Table 5 (emjo).
- We have corrected and ensured a consistent citation style in Reference (ada6).
Please don't hesitate to let us know if you have any additional comments on the revised manuscript.

[...continuing...]

---

> ### Author Response · Authors · 2025-11-24
> **General Response (2/2)**
>
> ## Probing Experiment: Empirical Validation of Semantic-Acoustic Modeling
>
> To address common concerns on interpretability and verify our claims, we conducted **Layer-wise Probing Experiments** using simple linear classifiers trained on the frozen internal hidden states of VoxCPM following the SUPERB [R1] setting. The  goal was to analyze the specialization of our hierarchical modules on linguistic content (Automatic Speech Recognition and Phoneme Recognition, ASR & PR) versus acoustic identity (Automatic Speech Recognition, ASV).
>
> The performance metrics used are: Word Error Rate (WER) and Phoneme Error Rate (PER), where lower values indicate better linguistic content preservation (higher intelligibility); and Equal Error Rate (EER), where lower values indicate better speaker identity modeling (higher acoustic/timbre distinguishability).
>
> The probing experiment results offer evidence to our core architectural claims:
> - **Acoustic Features Aggregation:** The LocEnc is responsible for aggregating the low-level continuous acoustic latents from AudioVAE. Consequently, it reflects the weak semantic relevance (PER=59.12%, WER=65.79%) and carries fundamental acoustic information like timbre (EER=15.38%).
> - **Semantic Specialization**: The TSLM representation achieves the lowest PER (45.60%) and WER (60.43%), validating its role in capturing the stable linguistic and prosodic content.
> - **Acoustic Filtering**: The FSQ representation exhibits the highest EER (19.25%). This is a critical finding: the quantization bottleneck effectively acts as an acoustic filter, actively reducing correlation with speaker identity and prioritizing content-prosodic stability.
> - **Residual Recovery**: Conversely, the RALM representation achieves the lowest EER (13.24%) and relatively high WER (64.85%) and PER (53.49%). This conclusively proves that the RALM specializes in recovering the fine-grained acoustic residuals and speaker identity that were filtered by the FSQ.
>
> The slight rise in content error post-FSQ (PER=50.90%, WER=62.37%) confirms the expected lossy nature of quantization and the subsequent shift in the RALM's focus from pure semantic modeling toward acoustic refinement. These findings empirically validate our premise: VoxCPM successfully enforces a natural division of labor using the FSQ as an inductive bias.
>
> Table 1: Layer-wise probing results on internal hidden states of VoxCPM.
> | Hidden State Location  | PR (PER) ↓ | ASR (WER) ↓ | ASV (EER) ↓ |
> |------------------|------------|-------------|-------------|
> | LocEnc output  | 59.12      | 65.79  | 15.38   |
> | TSLM last hidden (Pre-FSQ)  | **45.60**      | **60.43** | 18.70  |
> | FSQ output | 50.90      | 62.37   | 19.25 |
> | RALM last hidden | 53.49      | 64.85   | **13.24**  |
>
> >[R1] Yang, Shu-wen, et al. "SUPERB: Speech Processing Universal PERformance Benchmark." Interspeech 2021 (2021).

---

### Meta-Review · Area_Chair_A4g1 · 2026-01-09

**Summary:**

The paper presents VoxCPM, a hierarchical TTS framework using a semi-discrete FSQ bottleneck to separate semantic andacoustic tasks., While the concept is interesting, the decision to reject is primarily informed by the following major concems:System Complexity and Dependency: The pipeline involves multiple interacting components (LocEnc, TSLM, RALM, LocDiT)which, despite gradient flow, increases the dificulty of scaling and robust fine-tuning. The reliance on a pre-trained frozenAudioVAE limits the "true" end-to-end nature and poses an information bottleneck.haok of Toranspdtefoery'ePerfortmnane Ceainesi otere Mfe ely, PC,e rolianbou, tratfe da tasetumating te SC Tfsleaitc sdata compared to existing more efficient models.Potential Instability: The lack of explicit alignment mechanisms raises significant concerns regarding timing stability andmispronunciations in real-world,long-form synthesis. Mixed results in subjective naturainess also suggest the model mayprioritize speaker similarity over general speech quality.

**Reviewer Concerns:**

Addressed:
Efficiency and Latency: The authors successfully provided RTF (O.17) and throughput data showing real-time viability.Interpretability of FSQ: Probing experiments (PERMER/EER) provided empirical evidence that TSLM captures content whileRALM captures identity.
Outstanding:
Reproducibility of SOT A: The dependency on the 1M-hour private dataset remains a fundamental issue for academic verification.Explicit Alignment: The authors' reliance on implicit alignment is a design choice that still leaves the model vuinerable to prosodicdrift and instability in complex or long-form scenarios.Naturalness Ceiling: The mixed DNSMOS and naturaliness results indicate that the current architecture may struggle to generatehigh-quality speech when prompt audio is suboptimal, lacking "enhancement" capabilities found in competitors.

**Reviewer Scores:**

Reviewer Nviv (Current: 8): Likely remain at 8. They were satisfied with the additional experiments.Reviewer ada6(Current: 6): Likely stay at 6. While the theoretical 1Bgrounding was added, the reproducibility concern regardingthe 1M-hour dataset is a hard constraint.
Reviewer emjo (Current: 4): Might increase to 5. The FSQ vs. VAE ablation addressed their core technical doubt, but theend-to-end definition remains a point of contention.Reviewer KJR8 (Current: 4): Likely stay at 4. The rebuttal admitted the potential for timing instability and didn't introduce explicitalignment mechanisms, which was their main concern.Reviewer MsWh (Current: 4): Likely stay at 4. The "Faithful Cloning" explanation for negative naturainess results is unlikely tochange their view on the model's overall performance as a cascaded vs. E2E system.

---

### Decision · Program_Chairs · 2026-01-26

Accept (Poster)